# Unique phenotypes and clonal expansions of human CD4 effector memory T cells re-expressing CD45RA

Yuan Tian[1], Mariana Babor[1], Jerome Lane[1], Veronique Schulten[1], Veena S. Patil[1], Grégory Seumois [1], Sandy L. Rosales[1], Zheng Fu [2], Gaelle Picarda[3], Julie Burel[1], Jose Zapardiel-Gonzalo[1], Rashika N. Tennekoon[1,4], Aruna D. De Silva[1,4], Sunil Premawansa[5], Gayani Premawansa[6], Ananda Wijewickrama[7], Jason A. Greenbaum[2], Pandurangan Vijayanand[1], Daniela Weiskopf[1], Alessandro Sette[1] & Bjoern Peters[1]

The expression of CD45RA is generally associated with naive T cells. However, a subset of effector memory T cells re-expresses CD45RA (termed TEMRA) after antigenic stimulation with unknown molecular characteristics and functions. CD4 TEMRA cells have been implicated in protective immunity against pathogens such as dengue virus (DENV). Here we show that not only the frequency but also the phenotype of CD4 TEMRA cells are heterogeneous between individuals. These cells can be subdivided into two major subsets based on the expression of the adhesion G protein-coupled receptor GPR56, and GPR56$^+$ TEMRA cells display a transcriptional and proteomic program with cytotoxic features that is distinct from effector memory T cells. Moreover, GPR56$^+$ TEMRA cells have higher levels of clonal expansion and contain the majority of virus-specific TEMRA cells. Overall, this study reveals the heterogeneity of CD4 TEMRA cells and provides insights into T-cell responses against DENV and other viral pathogens.

[1] Division of Vaccine Discovery, La Jolla Institute for Allergy and Immunology, La Jolla, CA 92037, USA. [2] Bioinformatics Core Facility, La Jolla Institute for Allergy and Immunology, La Jolla, CA 92037, USA. [3] Division of Immune Regulation, La Jolla Institute for Allergy and Immunology, La Jolla, CA 92037, USA. [4] Genetech Research Institute, Colombo 00800, Sri Lanka. [5] Department of Zoology and Environment Sciences, Science Faculty, University of Colombo, Colombo 00700, Sri Lanka. [6] North Colombo Teaching Hospital, Ragama 11010, Sri Lanka. [7] National Institute of Infectious Diseases, Gothatuwa, Angoda 10620, Sri Lanka. Yuan Tian and Mariana Babor contributed equally to this work. Correspondence and requests for materials should be addressed to A.S. (email: alex@lji.org) or to B.P. (email: bpeters@lji.org)

T cells have important functions in conferring immunological protection against infectious pathogens by generating effector cells that mediate antigen control and by forming memory cells that provide long-term protective immunity against recurring infections. Effector and memory T cells are diversified into distinct subsets with specialized functions, and numerous molecules have been used to help identify those subsets and characterize the heterogeneity of both CD4 and CD8 T cells[1]. On the basis of the expression of two surface molecules, CD45RA and CCR7, human T cells can be divided into four subsets, including CD45RA+CCR7+ naive (TN), CD45RA−CCR7+ central memory (TCM), CD45RA−CCR7− effector memory (TEM), and CD45RA+CCR7− effector memory re-expressing CD45RA (TEMRA) T cells[1,2]. TEMRA cells have mostly been studied in the CD8 T-cell compartment, where they are found at appreciable frequencies in most individuals[2–5]. By contrast, the frequency of CD4 TEMRA cells varies drastically between individuals ranging from <0.3% to nearly 18% of total CD4 T cells in an apparently healthy population[6], and their functional role is less clear. Accumulating studies have indicated that infections with pathogens such as human cytomegalovirus (CMV) and dengue virus (DENV) are associated with an expansion of CD4 TEMRA cells[7–9]. In addition to exhibiting a CD45RA+CCR7− phenotype, CD4 TEMRA cells have also been characterized by decreased expression of CD27 and CD28, as well as increased expressions of CD57 and effector molecules such as perforin and granzyme B that resemble more terminally differentiated state[5,9,10].

Studies of DENV-infected individuals suggested a functional significance of CD4 TEMRA cells[9]. It was shown that the frequency of CD4 TEMRA cells progressively expands as a function of DENV infection history[9]. CD4 TEMRA cells associated with this expansion have a cytotoxic phenotype and exhibit increased expression of the chemokine receptor CX3CR1, which is associated with both CD4 and CD8 T cells that possess cytotoxic potentials[9,11–13]. Moreover, enhanced magnitude and functionality of CD4 TEMRA cells correlate with HLA allelic variants that are associated with relative resistance to severe DENV diseases, suggesting that CD4 TEMRA cells may have a protective function in this setting[9,14]. Nevertheless, how CD4 TEMRA cells differ from other memory-phenotype CD4 T cells such as TCM and TEM cells at the global level is less well defined. Lastly, it remains to be addressed whether CD4 TEMRA cells represent a homogenous population, or heterogeneity exists within this subset.

In this study, we set out to comprehensively define the immune signatures of CD4 TEMRA cells. We find that CD4 TEMRA cells have highly diverse gene expression profiles in different donors. In some donors, TEMRA cells are similar to conventional TEM cells. However, in other donors, by comparison with their TCM and

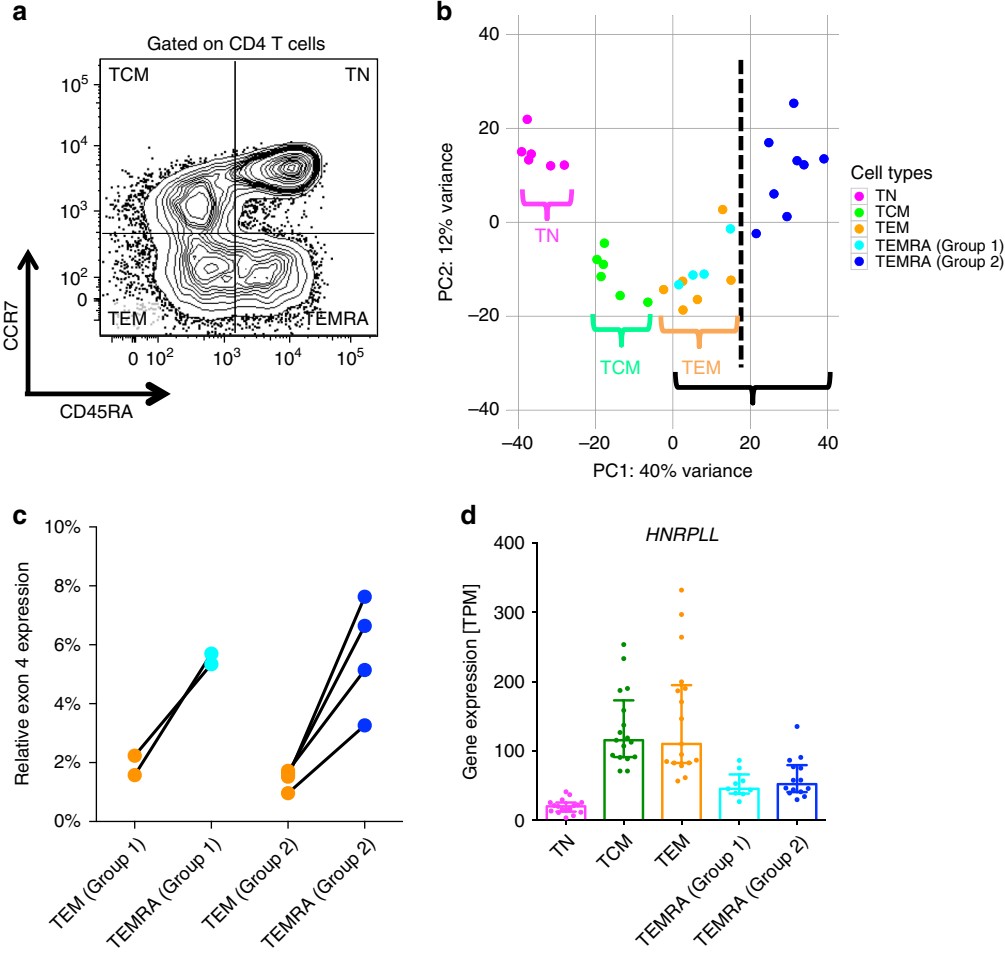

**Fig. 1** Gene expression data indicates that CD4 effector memory T cells re-expressing CD45RA (TEMRA) are highly variable between different donors. **a** Representative staining of the memory CD4 T-cell subsets. **b** PCA analysis of gene expression data (top 1000 variable genes) for different CD4 T-cell subsets (n = 6 for TN, TCM and TEM, and n = 12 for TEMRA). Note the dashed vertical line separates TEMRA from group 2 donors from all other cell subsets. **c** CD45 exon 4 abundance in TEMRA with respect to effector memory T (TEM) cells. The line connects TEM and TEMRA relative exon 4 expression for each individual donor (n = 2 and 4 for group 1 and group 2 donors, respectively). **d** The expression levels of HNRPLL in each of the different cell types (n = 17 for TN, TCM and TEM, and n = 9 and 14 for TEMRA cells from group 1 and group 2 donors, respectively). Error bars show median with interquartile range

TEM counterparts, TEMRA cells display a unique gene expression profile, which is characterized by the upregulation of cytotoxic molecules such as GPR56, CD244, perforin and granzyme B, as well as transcription factors such as Runx3, T-bet and Hobit. We show that this variability between donors is due to the presence of two primary sub-populations of TEMRA cells, with the TEM-like GPR56⁻ TEMRA subpopulation being present in all donors with similar frequency, while the cytotoxic GPR56⁺ TEMRA sub-population have high variability from donor to donor with evidence for clonal expansion. Furthermore, the majority of DENV-specific, as well as CMV- and Epstein–Barr virus (EBV)-specific CD4 TEMRA cells are found in the GPR56⁺ TEMRA subset. Thus, GPR56⁺ TEMRA cells may have an important function in the immune response against DENV and other viral pathogens.

## Results

**Gene expression profiles of CD4 TEMRA cells**. To better understand the phenotypic and functional characteristics of CD4 effector memory T cells re-expressing CD45RA (TEMRA) compared to other memory cell subsets, we isolated naive CD4 T cells (TN), as well as memory CD4 T-cell subsets, including central memory (TCM), effector memory (TEM), and TEMRA cells based on the expression of CCR7 and CD45RA (Fig. 1a and Supplementary Fig. 1a) for RNA-sequencing. Samples were obtained from 12 individuals from the Colombo region, Sri Lanka (Supplementary Table 1, cohort 1), including nine individuals that had been previously infected with DENV, which is hyperendemic in Colombo. Consistent with previous results[9], a wide variation of CD4 TEMRA cell frequencies were observed ranging from 0.4% to 18% of total CD4 T cells (Supplementary Table 1, cohort 1).

To visualize the global gene expression patterns of different T-cell memory subsets, we performed Principal components analysis (PCA). Interestingly, CD4 TN, TCM, and TEM samples all grouped into distinct clusters. In contrast, CD4 TEMRA cells were associated with a higher level of variation, with TEMRA cells from some individuals overlapping with TEM cells, whereas TEMRA cells from others were more separated from CD4 TEM cells (Fig. 1b). This trend was consistently observed independent of the number of genes included in the PCA (Supplementary Fig. 2a), suggesting that CD4 TEMRA cells isolated from different donors have distinct expression patterns. To explore this further, we performed an unbiased clustering analysis of TEMRA samples, which identified (at least) two major clusters of CD4 TEMRA cells (Supplementary Fig. 2b). When this classification was applied to the TEMRA samples in the original PCA plot as shown in Fig. 1b, two clusters of TEMRA samples were apparent (Supplementary Fig. 2c). Thus, according to TEMRA phenotypes, we classified the donors into group 1 and group 2, whose CD4 TEMRA cell gene expression profiles were similar to or distinct from TEM cells, respectively.

**Down-regulation of HNRPLL is associated with CD4 TEMRA cells**. Given the overlap of the gene expression profiles between CD4 TEMRA and TEM cells in group 1 donors, we were interested in finding genes that differentiate TEMRA and TEM cells. CD4 TEMRA cells express the CD45RA isoform, while TEM cells express CD45RO but lack CD45RA expression[1,9]. Transition from CD45RA to CD45RO implies the skipping of PTPRC (encodes CD45) exons 4, 5, and 6, a process controlled by the protein heterogeneous nuclear ribonucleoprotein L-like (HNRPLL)[15,16]. Indeed, differential expression analysis of using the DEXSeq package[17,18] showed that the expression of PTPRC exons 4, 5, and 6 was higher in TEMRA than TEM cells (Supplementary Fig. 3). Importantly, exon 4 (present in the CD45 isoform recognized by the CD45RA antibody) was more abundant in TEMRA cells than

in TEM cells for both group 1 and group 2 donors (Fig. 1c). Moreover, as shown in Fig. 1d, the expression of HNRPLL, which induces the transition of expressing CD45RA to CD45RO in activated T cells[16], was consistently higher in CD4 TEM and TCM cells compared with TEMRA cells, and no marked difference was observed between group 1 and group 2 donors. Taken together, these data excluded the possibility that the observed difference between TEMRA cells from group 1 and group 2 donors was a result of an ineffective cell sorting process.

To assess the reproducibility of our findings, an independent second cohort composed of 5 donors that had been previously infected with DENV from the same region (Colombo, Sri Lanka) was analyzed (Supplementary Table 1, cohort 2). Furthermore, to investigate if the dichotomous presence of TEMRA cells from group 1 and group 2 donors is a general phenomenon, we investigated a third cohort composed of 6 healthy individuals from San Diego (California, USA) that had not been exposed to DENV (Supplementary Table 1, cohort 3). Interestingly, we found both group 1 and group 2 donors in both of these cohorts (Supplementary Fig. 4), supporting the notion that the heterogeneity of CD4 TEMRA cell transcriptional profiles between individuals observed in our initial cohort is a general feature of this cellular subset.

**TEMRA cells from group 2 donors have distinct gene signatures**. We next examined which genes were differentially expressed between CD4 TEMRA cells from group 1 and group 2 donors, and how they differed from TEM cells. Figure 2a depicts a Venn diagram enumerating how many genes were differentially expressed (Padj < 0.05) with an effect size similar or greater to what we observed for the HNRPLL (fold change > 1.87), which can be used to distinguish any TEMRA cell from TEM as described above. Differential expression analyses for cohort 1 and cohorts 2, 3 separately are shown in Supplementary Fig. 5. Surprisingly, only four genes, including HNRPLL, were differentially expressed between CD4 TEM and CD4 TEMRA from group 1 donors (Fig. 2a), suggesting that CD4 TEMRA cells from group 1 donors had a largely similar transcriptional program compared to CD4 TEM cells. In contrast, the gene expression profiles of CD4 TEMRA cells from group 2 donors was considerably different, with 438 and 341 differentially expressed genes, when compared with TEM cells and TEMRA cell from group 1 donors, respectively. There were 228 genes specific for TEMRA cells from group 2 donors that were consistently upregulated or downregulated compared with both TEM cells and TEMRA cells from group 1 donors (Fig. 2a).

Next, we used Ingenuity Pathway Analysis (IPA) to functionally characterize TEMRA cells from group 2 donors. This enrichment analysis showed that cell migration and cytotoxicity were the top two functions when sorted by P-values. Moreover, the expression of genes associated with cell migration was significantly downregulated in TEMRA cells from group 2 donors ($p = 1.03E-17$ using right-tailed Fisher exact test, activation $z$-score = −2.108), while the expression of cytotoxicity related genes was significantly upregulated ($p = 2.41E-11$ using right-tailed Fisher exact test, activation $z$-score = 2.093) (Supplementary Tables 4 and 5), supporting the notion that TEMRA cells from group 2 donors exhibited a cytotoxic phenotype.

**Validation of TEMRA cell signatures in the protein level**. To validate the RNA expression signature of CD4 TEMRA cells from group 2 donors at the protein level, we performed fluorescence-based flow cytometry analysis on 11 selected markers that showed differential expression in transcriptomic analysis and for which antibody staining panels were available to us. By comparison with TEM cells and TEMRA cells from group 1 donors, GPR56

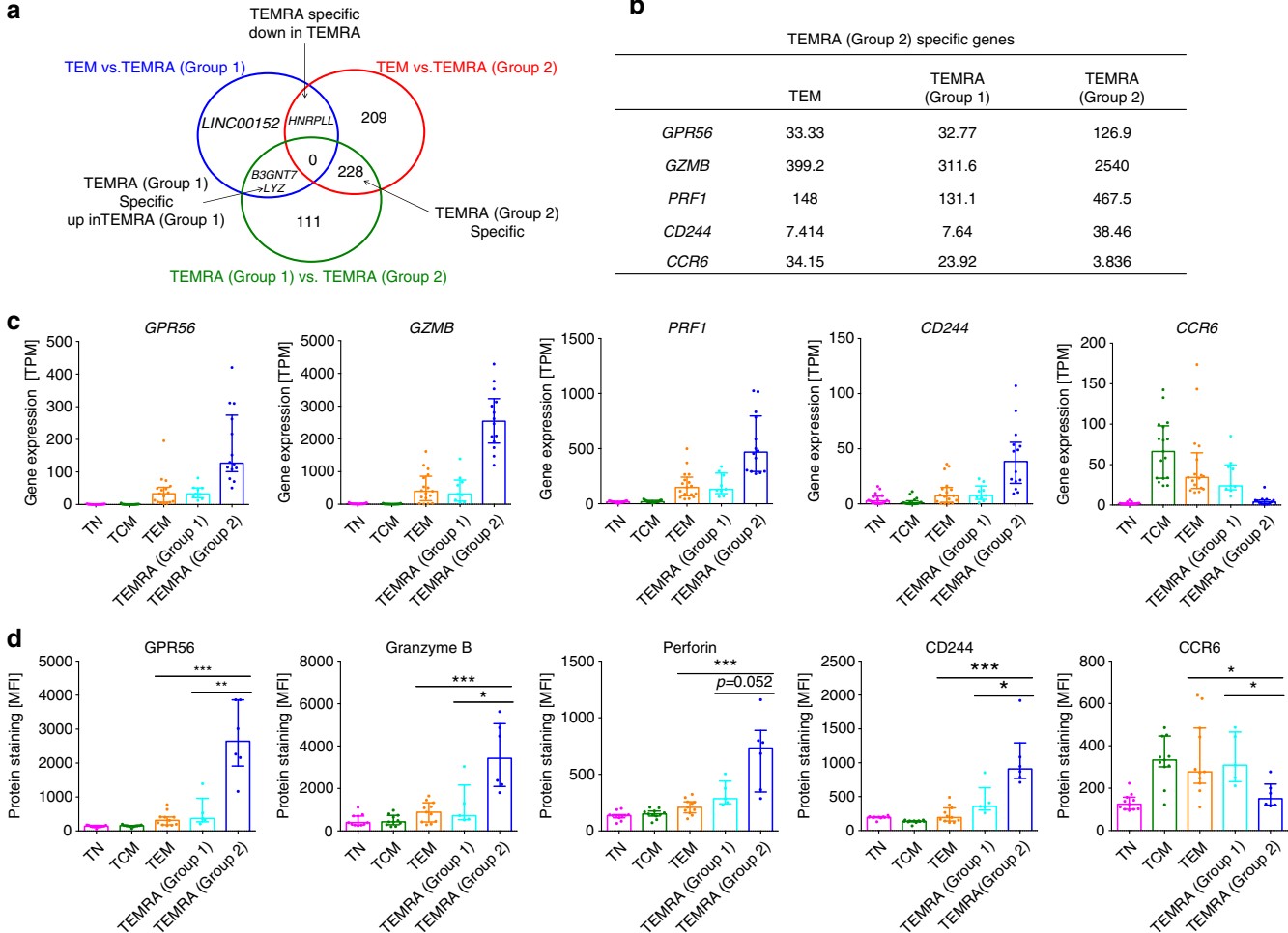

**Fig. 2** Identification of specific transcriptomic profiles for CD4 effector memory T cells re-expressing CD45RA (TEMRA) from group 1 and group 2 donors. **a** Pairwise differential expression analysis between CD4 effector memory T (TEM) cells, CD4 TEMRA cells from group 1 donors, and CD4 TEMRA cells from group 2 donors. A total of 228 genes that are specific for TEMRA cells from group 2 donors were identified ($n = 17$, 9 and 14 for TEM, TEMRA cells from group 1 donors and TEMRA cells from group 2 donors, respectively). **b** Transcriptomics median and Padj values for the five markers significantly upregulated/downregulated in TEMRA cells from group 2 donors. **c** Bar graphs show gene expression values in transcripts per million (TPM) for various molecules in different cell subsets ($n = 17$ for naive (TN), central memory (TCM) and TEM, and $n = 9$ and 14 for TEMRA cells from group 1 and group 2 donors, respectively). **d** Bar graphs show protein abundance in mean fluorescence intensity (MFI) for various molecules in different cell subsets ($n = 11$ for TN, TCM and TEM, and $n = 5$ and 6 for TEMRA cells from group 1 and group 2 donors, respectively). Error bars show median with interquartile range. Statistical significance was determined by two-tailed Mann–Whitney test. *$p < 0.05$, **$p < 0.01$, ***$p < 0.001$

(encoded by *GPR56*), granzyme B (encoded by *GZMB*), perforin (encoded by *PRF1*), and CD244 (encoded by *CD244*), all related to cytotoxicity based on the IPA analysis, showed substantially higher expression levels in TEMRA cells from group 2 donors at both mRNA (Fig. 2b, c) and protein (Fig. 2d) levels, while the expression levels for the migration marker CCR6 (encoded by *CCR6*) were reduced in TEMRA from group 2 donors (Fig. 2b–d). Of the remaining 6 selected markers, CX3CR1 and T-bet showed significant differences between TEMRA cells from group 2 donors and TEM cells but not between TEMRA cells from group 2 donors and TERMA cells from group 1 donors, and the rest four markers showed no statistically significant differences (TNFRSF10A, TCF1, KLRG1 and granulysin, which are encoded by *TNFRSF10A*, *TCF7*, *KLRG1* and *GNLY*, respectively) (Supplementary Fig. 6). Thus, a substantial proportion of the markers identified based on mRNA expression also showed substantial differences in their protein expression in TEMRA cells from group 2 donors.

Given the cytotoxic nature of the gene and protein signatures of TEMRA cells from group 2 donors, we further evaluated the expression of three transcription factors that have

been implicated in regulating the cytotoxic program in CD4 T cells, including ThPOK, Runx3 and Hobit[14,19]. Although the genes encoding these transcription factors did not reach statistical significance in our gene expression comparisons, we observed a trend of higher expression of *ZNF683*, which encodes Hobit, in TEMRA cells from group 2 donors (Fig. 3a). At the protein level, TEMRA cells from group 2 donors upregulated the expression of Runx3 while exhibiting reduced expression of ThPOK (Fig. 3b), which are two transcription factors that reciprocally regulate the differentiation of cytotoxic CD4 T cells[14]. Moreover, TEMRA cells from group 2 donors had significantly enhanced protein expression of Hobit (Fig. 3b), which is upregulated in human effector-phenotype CD8 T cells[20] and has been shown to distinguish human cytotoxic CD4 T cells[19]. Thus, TEMRA cells from group 2 donors exhibited a transcriptional signature that favors the programming of cytotoxic CD4 T cells.

Next, we asked if the diversity in protein expression among CD4 TEMRA cells observed at the level of cell populations derived from different donors was a consequence of a global shift

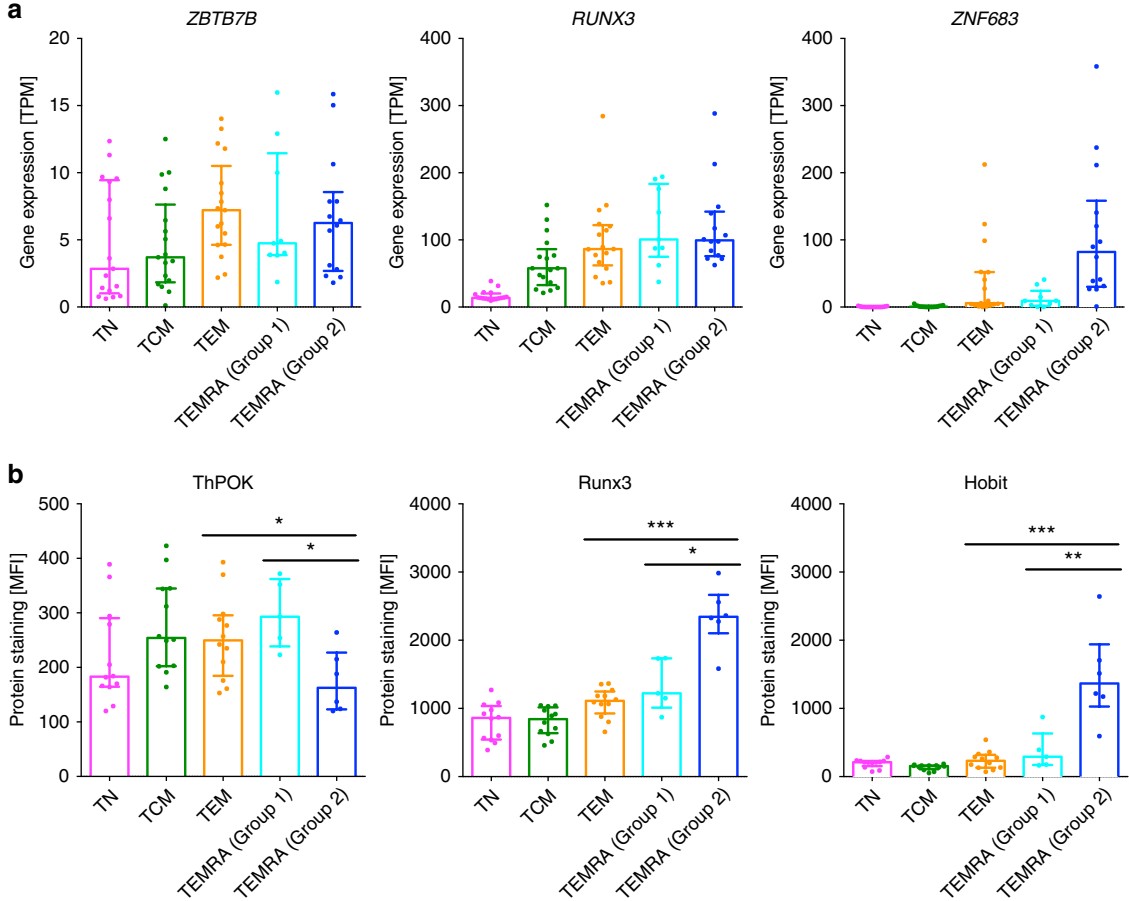

**Fig. 3** TEMRA from group 2 donors exhibit differential expression of key transcriptional factors that regulate the development of cytotoxic CD4 T cells in the protein level. **a** Bar graphs show gene expression values in transcripts per million (TPM) for *ZBTB7B*, *RUNX3* and *ZNF683* in different cell subsets ($n = 17$ for TN, TCM and TEM, and $n = 9$ and 14 for TEMRA cells from group 1 and group 2 donors, respectively). **b** Bar graphs show protein abundance in mean fluorescence intensity (MFI) for ThPOK, Runx3, and Hobit in different cell subsets ($n = 11$ for TN, TCM and TEM, and $n = 5$ and 6 for TEMRA cells from group 1 and group 2 donors, respectively). Error bars show median with interquartile range. Statistical significance was determined by two-tailed Mann–Whitney test. *$p < 0.05$, **$p < 0.01$, ***$p < 0.001$

in the protein expression profile in all cells, or if it is reflective of a shift in the subset composition of TEMRA cells between donors. To address this question, we measured the abundance of GPR56 and perforin, two markers specifically upregulated in TEMRA cells from group 2 donors. Intriguingly, by comparison with TEMRA cells from group 1 donors, a greater fraction of TEMRA cells from group 2 donors were GPR56+Perforin+ (Fig. 4a, b and Supplementary Fig. 1b), suggesting that rather than a global shift in expression levels, there are distinct subsets of TEMRA cells found in each donor. Figure 4c shows the frequencies of GPR56−Perforin− and GPR56+Perforin+ TEMRA cells as a fraction of total CD4 T cells for all the eleven donors analyzed (Supplementary Table 1, cohorts 2-3). Interestingly, the frequency of GPR56−Perforin− TEMRA cells was low and nearly constant in both group 1 and group 2 donors. In contrast, the frequency of GPR56+Perforin+ TEMRA cells was very low in group 1 donors, but was substantially higher in group 2 donors.

On the basis of this observation, we wanted to determine if the observed differences in total CD4 TEMRA cells between donors could be explained based on a constant fraction of GPR56−Perforin− cells, with a variable component of TEMRA cells positive for GPR56 or Perforin. Indeed, a simple model assuming a constant fraction of 0.8% GPR56−Perforin− cells plus a variable fraction of TEMRA cells positive for GPR56 or Perforin explained the relationship between the overall frequency of

TEMRA cells and the frequency of GPR56−Perforin− TEMRA cells (Fig. 4d). This model suggests that donors with a low frequency of CD4 TEMRA cells tend to have a majority of GPR56−Perforin− TEMRA cells. On the other hand, as the frequency of the TEMRA population within CD4 T cells increases, the proportion of GPR56−Perforin− cells within the TEMRA population is considerably reduced. Thus, group 2 donors, who tended to have a higher TEMRA frequency, had mainly TEMRA cells positive for GPR56 or perforin (Fig. 4a–c).

**CyTOF analysis reveals CD4 TEMRA cell differentiation states.** To further investigate the heterogeneity of CD4 TEMRA cells, we analyzed the expression of 21 proteins by CD4 TEMRA cells simultaneously at a single-cell level using cytometry by time-of-flight (CyTOF). We selected four donors, two with higher frequencies of CD4 TEMRA cells and two with lower frequencies. CD4 TEMRA cells were gated as CD14−CD19−CD3+CD8−CD4+CD45RA+CCR7− (Fig. 5a). The 21-marker profile of the gated TEMRA cells was visualized using viSNE, which employs t-stochastic neighbor embedding (t-SNE) to generate a two-dimensional map where the distance between cells corresponds to their marker profile similarity[21]. The resulting viSNE maps revealed several distinct islands of CD4 TEMRA cells (Fig. 5b) with two main islands corresponding to the GPR56+ or

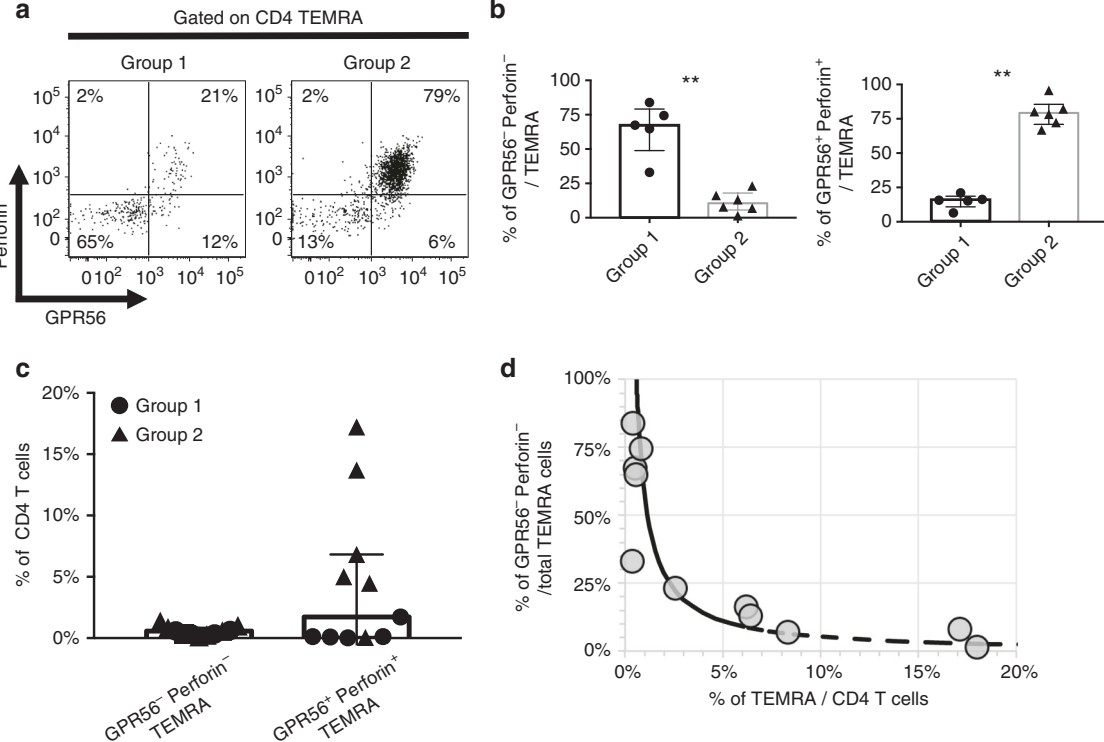

**Fig. 4** Identification of TEMRA subtypes within a given individual. **a** Representative flow cytometry plots show the expression of GPR56 and perforin by CD4 TEMRA cells in group 1 and group 2 donors. **b** Bar graphs show the percentages of GPR56⁻perforin⁻ (left panel) and GPR56⁺perforin⁺ (right panel) TEMRA cells in group 1 and group 2 donors ($n = 5$ and 9 or group 1 and group 2 donors, respectively). Error bars show median with interquartile range. Statistical significance was determined by two-tailed Mann–Whitney test. **p < 0.01. **c** Plot shows the percentages of GPR56⁻perforin⁻ and GPR56⁺perforin⁺ TEMRA cells with respect to the total CD4 T cells in group 1 and group 2 donors ($n = 11$). Error bars show median with interquartile range. **d** Plot shows that the proportion of TEMRA cells that were GPR56⁻perforin⁻ within the TEMRA population decreases exponentially as the TEMRA frequency within the CD4 T cells increases ($n = 11$)

GPR56⁻ subsets as described above. CD4 TEMRA cells in donors with lower TEMRA frequencies (donors 1 and 2) fell nearly exclusively in the GPR56-negative island that also lacked the expression of KLRG1, CD244, granzyme B, T-bet, Runx3, and perforin (Fig. 5b). In contrast, cells in donors with higher frequencies of TEMRA cells (donors 3 and 4) displayed a substantial number of cells in the GPR56⁺ island that showed enhanced expression of KLRG1, CD244, granzyme B, T-bet, Runx3, and perforin (Fig. 5b). Interestingly, the other markers upregulated within this island showed distinct expression patterns. For example, granzyme B exhibited a pattern of gradient expression among TEMRA cells and was highly upregulated in a proportion of TEMRA cells, suggesting that CD4 TEMRA cells can be further divided into different states (Fig. 5c). Taken together, these data suggest that CD4 TEMRA cells contain at least two major subsets (GPR56⁻ and GPR56⁺) and that GPR56⁺ TEMRA cells are heterogeneous with distinct phenotypic attributes and their relative proportions vary between different donors.

**GPR56⁺ TEMRA cells have higher clonality than other subsets.** If the highly variable frequency of CD4 TEMRA cells in different donors is the outcome of differential clonal expansion events, then CD4 TEMRA cells from group 2 donors should have a more restricted TCR repertoire with a large number of cells sharing the same TCR clonotype. To test this hypothesis, we assembled TCR sequences from our RNA-Seq data, and compared the diversity of the assembled repertoire between different cell types, which was measured by normalized clonality (Eqs. 1 and 2) that indicates the extent to which one or a few TCR sequences dominate the

sample repertoire. Thus, smaller values of normalized clonality indicate a more polyclonal sample. Indeed, TEMRA cells from group 2 donors, which were enriched in GPR56⁺ TEMRA cells, displayed the highest clonality (0.375) compared to TEMRA cells from group 1 donors (0.09), TEM (0.05), TCM (0.01), and TN (0.01) cells, and this difference was statistically significant ($p < 0.001$, Mann–Whitney) in all comparisons (Fig. 6a and Supplementary Table 6). These data suggest that TEMRA cells from group 2 donors underwent more clonal expansion and thus had the least diverse TCR repertoire.

To examine the TCR repertoire of TEMRA cells in greater detail, we performed targeted TCR sequencing on additional samples from two donors with high TEMRA frequencies. Cells were sorted into TCM, TEM, GPR56⁻ TEMRA, and GPR56⁺ TEMRA subsets using the expression of GPR56 as a marker of the highly expanded TEMRA subset as described above. DNA was extracted and TCR-beta repertoire sequencing performed using the immunoSEQ assay from Adaptive Biotechnologies (Supplementary Table 7). To correct for differences in the number of T cells sequenced per cell type, we normalized each data set to 1000 TCRs using bootstrapping with replacement. Figure 6b shows the clonality of each cell type in the two donors. This plot suggests that GPR56⁺ TEMRA cells are responsible for the enhanced clonality of TEMRA cells from group 2 donors, as they exhibited much higher clonality compared with GPR56⁻ TEMRA cells, which had lower clonality than TEM cells.

To further assess possible relatedness of GPR56⁺ TEMRA cells and other memory cell types, we estimated the fraction of GPR56⁺ TEMRA clonotypes that were also present in either TEM

or TCM cells. Figure 6c shows that, on average, about 40% of the GPR56+ TEMRA clonotypes were present in TEM cells, although these clonotypes were rare in TEM cells, and this percentage increased to ~55% if the analysis was restricted to highly

represented clonotypes that had an abundance of at least 10 cells. Moreover, the highly represented clonotypes constituted about 80% of the GPR56+ TEMRA cells. Note that this trend was not observed for GPR56− TEMRA cells (Fig. 6c). On the other hand,

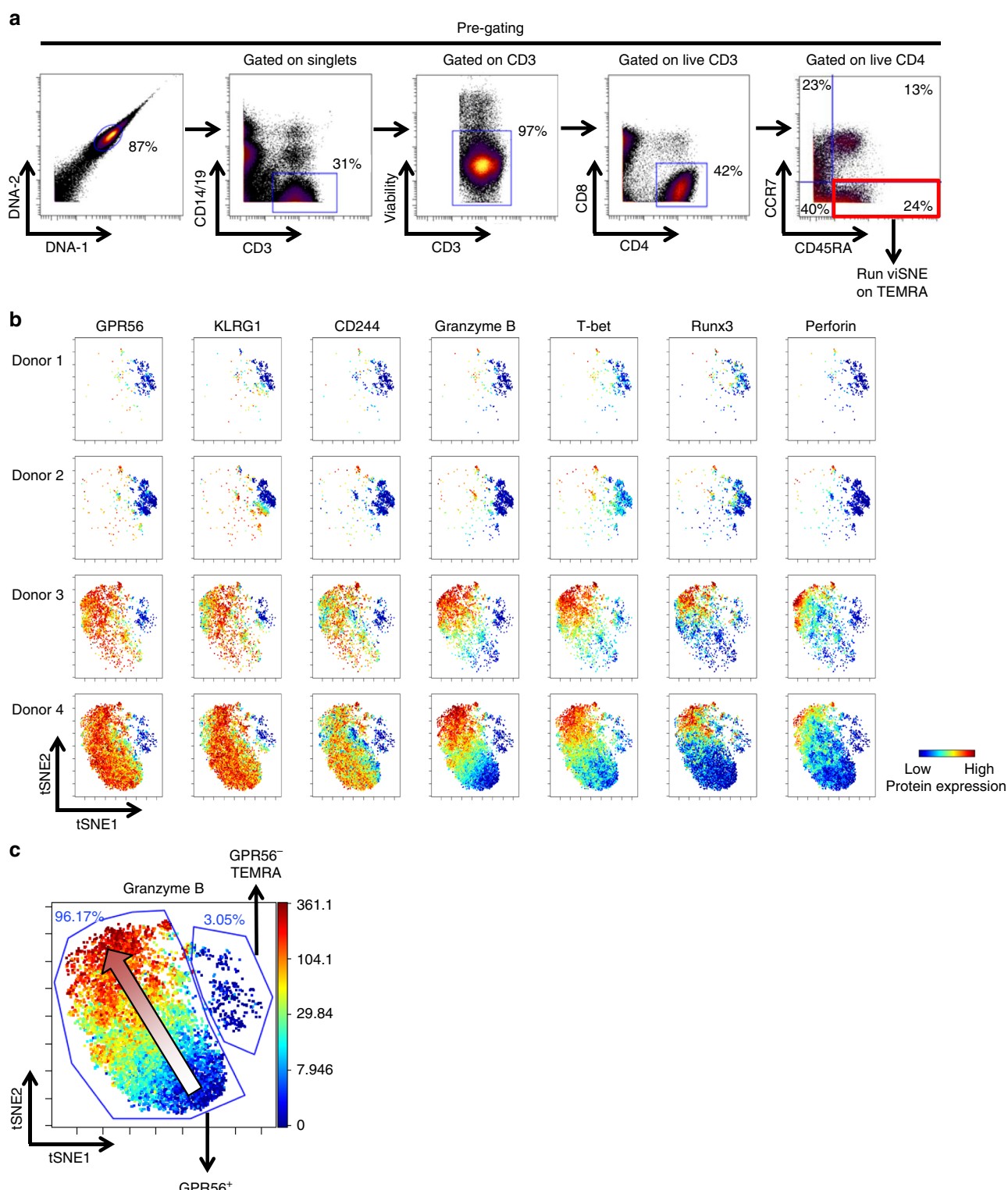

**Fig. 5** High dimensional CyTOF analysis reveals that TEMRA are highly heterogeneous. **a** Gating strategy for the identification of CD4 TEMRA cells (CD14−CD19−CD3+CD8−CD4+CD45RA+CCR7−). **b** viSNE analysis arranged cells along tSNE1 and tSNE 2 axes based on the expression of 21 proteins. Plots show the expression of GPR56, KLRG1, CD244, granzyme B, T-bet, Runx3, and perforin in each individual cell (n = 4). **c** Representative manual gating of GPR56− and GPR56+ TEMRA cells on a viSNE plot showing the expression of granzyme B

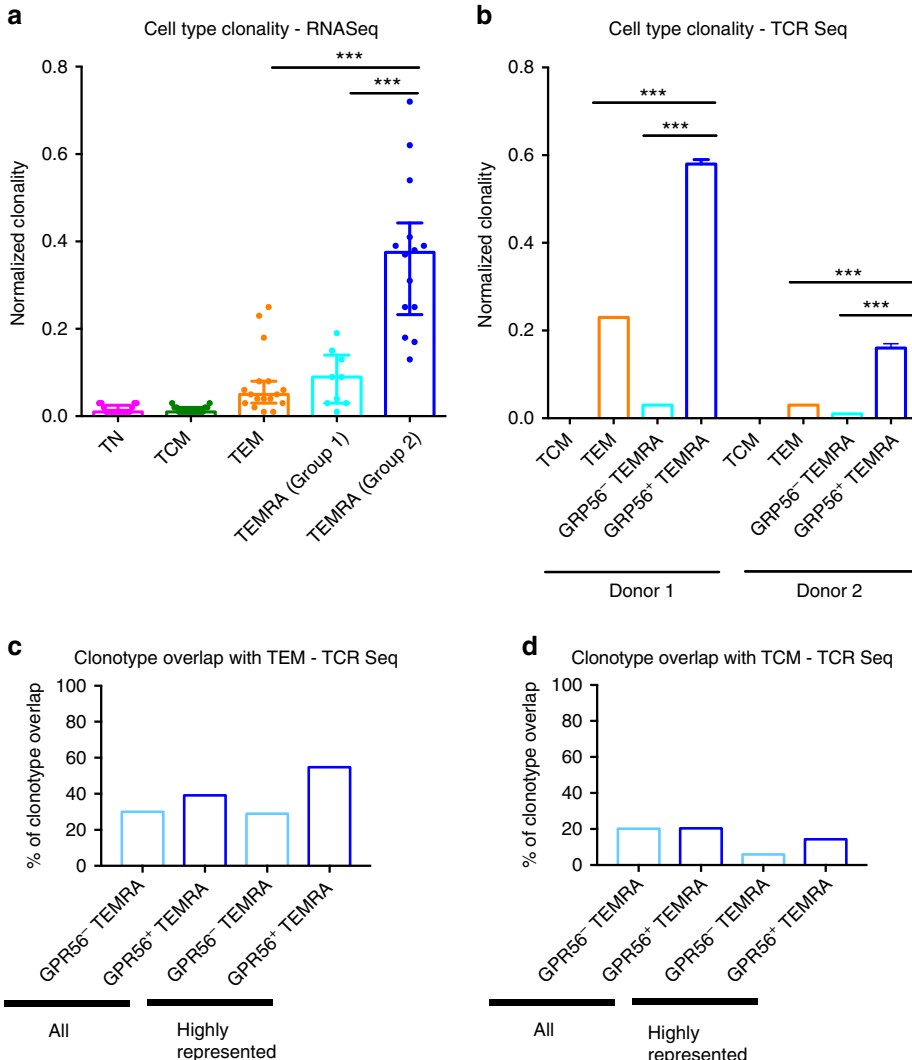

**Fig. 6** CD4 GPR56⁺ TEMRA display increased clonality. **a** Bar graph shows the normalized clonality of various CD4 T-cell subsets ($n = 17$ for naive (TN), central memory (TCM) and effector memory (TEM), and $n = 9$ and 14 for TEMRA cells from group 1 and group 2 donors, respectively). Error bars show median with interquartile range. Statistical significance was determined by two-tailed Mann–Whitney test. ***$p < 0.001$. **b** Bar graph shows the normalized clonality of GPR56⁺ TEMRA and GPR56⁻ TEMRA as well as TCM and TEM cells ($n = 2$) based on TCR-beta repertoire sequencing results. Error bars show median with interquartile range. Statistical significance was determined by two-tailed Mann–Whitney test. ***$p < 0.001$. **c**, **d** Bar graphs show the percentages of GPR56⁻ TEMRA and GPR56⁺ TEMRA clonotypes that overlapped with **c** TEM and **d** TCM cells, respectively ($n = 2$). A clonotype was defined as highly represented if it was presented in at least 10 cells. This threshold was determined to have roughly the top percentile clonotypes (>90%)

the overlap between GPR56⁺ TEMRA clonotypes and TCM cells was considerably lower (Fig. 6d). Thus, this analysis suggests that TEM cells are more likely to be a precursor of GPR56⁺ TEMRA cells than TCM cells, and GPR56⁺ TEMRA cells may result from clonal expansion of a subset of TEM cells. We cannot exclude the possibility that additional intermediate cell subsets may exist.

**DENV-specific CD4 TEMRA cells are predominantly GPR56⁺.**
Since previous findings indicated that DENV-specific CD4 T cells upregulate the expression of cytotoxic molecules in donors determined to have previous secondary DENV infections[9], we investigated whether DENV-specific CD4 TEMRA cells are predominantly found in the clonally expanded GPR56⁺ TEMRA cells. We identified DENV-specific TEMRA cells based upon their expression of IFN-γ after stimulation with DENV CD4 T-cell epitopes megapool (Supplementary Fig. 1b), and the frequencies of IFN-γ⁺ TEMRA cells ranged from 0.1 to 1.9% among a cohort of 10 donors associated with secondary DENV infections (Fig. 7a).

Please note that these samples were obtained anonymously from National Blood Center, Ministry of Health, Colombo, Sri Lanka, and the donors were healthy at the time of sample collection. We determined that these donors had secondary DENV infections by DENV-specific IgG ELISAs and flow cytometry-based neutralization assays (see Methods section). As shown in Fig. 7b, c, the majority of IFN-γ-producing TEMRA cells were GPR56⁺. Moreover, by comparison with total CD4 TEMRA cells, a larger proportion of IFN-γ-producing TEMRA cells co-expressed CD244 and perforin (Fig. 7b, c). To further confirm that GPR56⁺ TEMRA cells were major components of IFN-γ-producing cells in response to DENV, we sorted GPR56⁺ and GRP56⁻ TEMRA cells from secondary DENV-infected donors (Supplementary Fig. 1c) and measured their production of IFN-γ after stimulation with DENV CD4 T-cell megapool. We observed that a higher proportion of GRP56⁺ TERMA cells produced IFN-γ compared with their GRP56⁻ counterparts (Fig. 7d), further supporting the notion that DENV-specific TEMRA cells predominantly adopted a GPR56⁺ phenotype. Taken together, these data suggest that DENV-specific

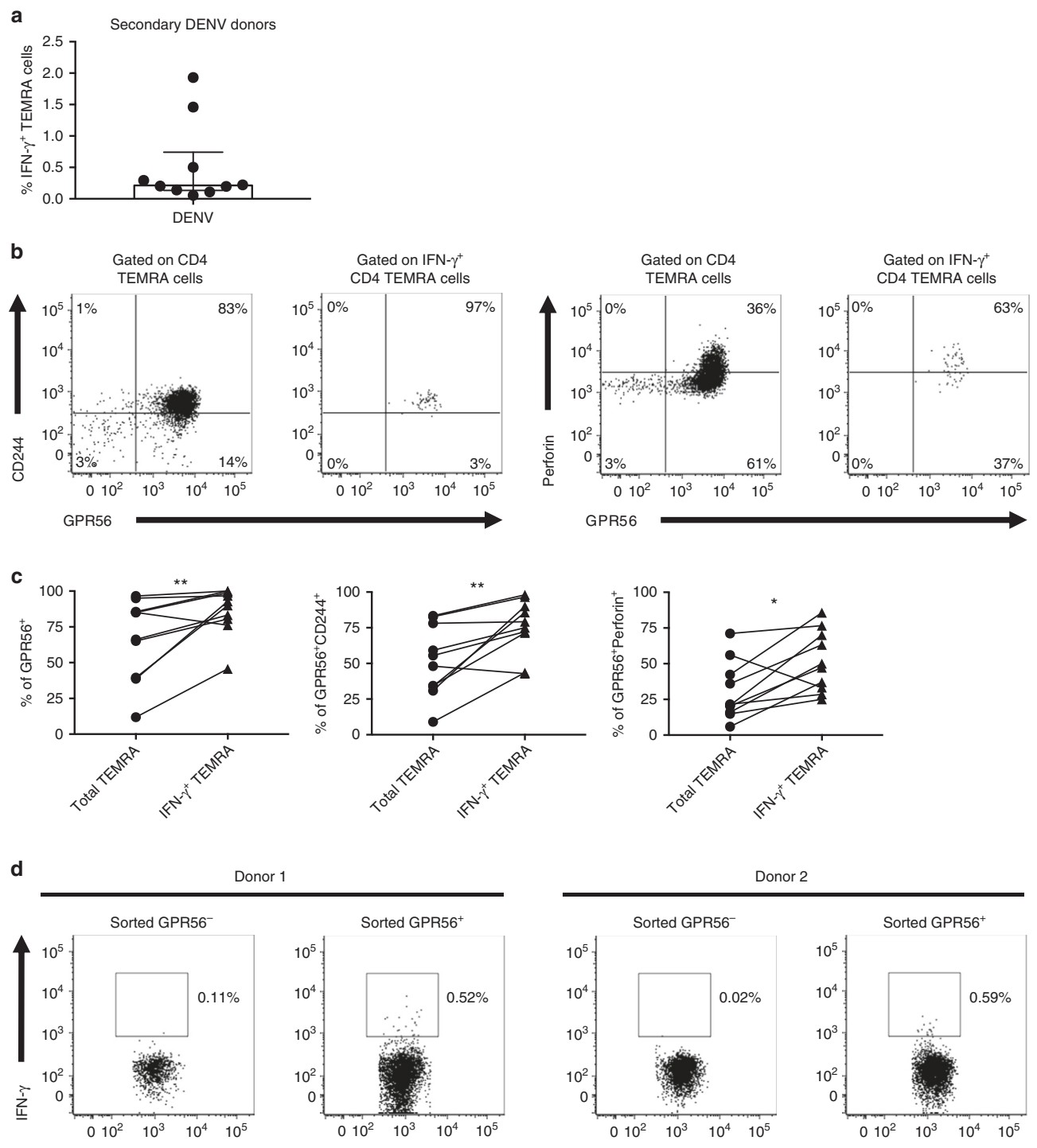

**Fig. 7** DENV-specific CD4 TEMRA are predominantly GPR56[+]. DENV-specific TEMRA cells were identified by the production of IFN-γ after stimulation with DENV CD4 megapool in donors associated with secondary DENV infections. **a** Bar graph shows the percentages of IFN-γ-producing CD4 TEMRA cells ($n = 10$). Error bars show median with interquartile range. **b** Flow cytometry plots show the expression of GPR56 and CD244 (left panel) or perforin (right panel) by total TEMRA cells and IFN-γ[+] TEMRA cells. **c** Graphs show the percentages of total and IFN-γ[+] GPR56[+] (left panel), GPR56[+]CD244[+] (middle panel), or GPR56[+]perforin[+] (right panel) TEMRA cells ($n = 10$). Statistical significance was determined by two-tailed Wilcoxon test. *$p < 0.05$, **$p < 0.01$. (**d**) Flow cytometry plots show the percentages of IFN-γ-producing cells among sorted GPR56[−] and GPR56[+] TEMRA cells ($n = 2$)

TEMRA cells are associated with a cytotoxic phenotype characterized by the expression of markers such as CD244, perforin, and especially GPR56. This is consistent with the observation that these cells were clonally expanded, which could result from multiple rounds of DENV infections.

**CMV- and EBV-specific TEMRA cells upregulate GPR56.** To investigate whether the observed phenotype of CD4 TEMRA cells specific to DENV was also found for other viruses characterized by multiple rounds of infection, we analyzed CMV- and EBV-specific CD4 TEMRA cells in a cohort of donors that had not

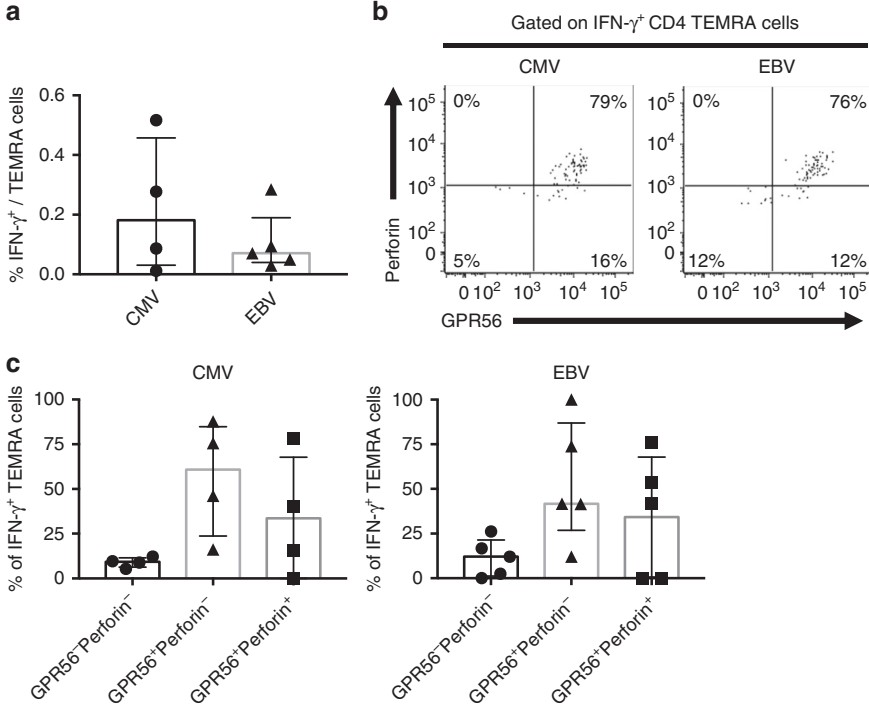

**Fig. 8** CMV- specific and EBV-specific CD4 TEMRA display heterogeneity and enhanced expression of GPR56. CMV- and EBV-specific TEMRA cells were identified by the production of IFN-γ after stimulation with CMV and EBV CD4 T-cell peptide pools, respectively. **a** Bar graph shows the percentages of IFN-γ-producing CD4 TEMRA cells (n = 4 and 5 for CMV and EBV, respectively). Error bars show median with interquartile range. **b** Flow cytometry plots show the expression of GPR56 and perforin by CMV- and EBV-specific TEMRA cells. **c** Graphs show the percentages of GPR56⁻Perforin⁻, GPR56⁺Perforin⁻, and GPR56⁺Perforin⁺ cells for CMV- (left panel) or EBV-specific (right panel) TEMRA cells (n = 4 and 5 for CMV and EBV, respectively). Error bars show median with interquartile range

been exposed to DENV. CMV- and EBV-specific CD4 TEMRA cells were detected by the production of IFN-γ after stimulation with CMV and EBV peptide pools, respectively (Fig. 8a and Supplementary Fig. 1b). Intriguingly, we observed that CMV-specific and EBV-specific CD4 TEMRA cells again predominantly displayed a GPR56⁺ phenotype (Fig. 8b, c), which was similar to the results of DENV-specific CD4 TEMRA cells as shown in Fig. 7. Thus, these data indicate that multiple viruses can elicit an expanded cytotoxic TEMRA population. This makes it likely that the presence of an expanded GPR56⁺ TEMRA subset might be reflective of the donor's infection history.

Finally, we found that in the case of DENV-, CMV-, and EBV-specific cells, TEMRA cells constituted only 9.8, 9.9 and 3.0% (median values) of the total response, respectively (Supplementary Fig. 7a). However, we observed that the TEMRA subset constituted for 34.1, 22.6, and 24.2% (median values) of the DENV-, CMV-, and EBV-specific cells that are associated with a cytotoxic phenotype (GPR56⁺Perforin⁺), respectively (Supplementary Fig. 7b), suggesting that the TEMRA subset is a relatively minor component of the response, but associated with a specific set of biological phenotypes.

## Discussion

CD4 effector memory T cells re-expressing CD45RA (TEMRA) have the greatest degree of variation in their frequency between donors in the general population[6]. Our study provides insights into the associations of this variability and the functional role of these cells. First, we showed that CD4 TEMRA cells are readily discernable from naive (TN) and other memory CD4 T-cell subsets including central memory (TCM) and effector memory (TEM) cells based on expression of CD45RA, CCR7, and *HNRPLL*. Second, we showed that the gene expression signatures

of CD4 TEMRA cells vary greatly between donors, with some displaying a profile similar to CD4 TEM cells, while others are clearly different. We demonstrated that this difference in TEMRA phenotype between donors was due to the difference in the frequency of GPR56⁺ TEMRA cells, which are dominant in some donors but not in others. GPR56⁺ TEMRA cells were characterized by the upregulation of various effector molecules and transcription factors that are involved in CD4 T-cell differentiation, cytotoxicity and migration. Third, we showed that GPR56⁺ TEMRA cells exhibit evidence for clonal expansion above that found in any other memory CD4 T-cell subset. Finally, we observed that the majority of IFN-γ-producing CD4 TEMRA cells in response to DENV, as well as CMV and EBV epitopes were found in the GPR56⁺ TEMRA subset. Taken together, these results suggest that there is a subset of clonally expanded cytotoxic CD4 TEMRA cells that is found in some but not all individuals.

In this study, we found that CD4 TEMRA cells from group 1 and group 2 donors exhibited distinct gene expression profiles, which was at least partially accounted for by the relative abundance of GPR56⁺ TEMRA cells. We observed a substantial expansion of GPR56⁺ TEMRA cells in group 2 donors, while the frequency of GPR56⁻ TEMRA cells within total CD4 T cells remained quite stable across the cohorts. One of the striking characteristics of GPR56⁺ TEMRA cells is the enhanced expression of molecules associated with cytotoxic activities such as CD244, granzyme B, perforin, and especially GPR56. Previous studies have shown that GPR56, with its closely related GPR97 and GPR114, is expressed on human cytotoxic lymphocytes such as natural killer cells and T cells, and may regulate their migratory properties[22]. In this study, we further demonstrate that GPR56 distinguishes CD4 TEMRA subsets with GPR56⁺ TEMRA cells

exhibiting enhanced expression of cytotoxic molecules, which is consistent with the previous findings[22]. Moreover, we were able to characterize the heterogeneity of CD4 TEMRA cells at an unprecedented level using cytometry by time-of-flight (CyTOF). While GPR56⁻ TEMRA cells uniformly displayed similar expression of the tested markers, GPR56⁺ TEMRA cells exhibited distinct expression patterns of the various makers. For example, KLRG1 and CD244 showed broad upregulation on GPR56⁺ TEMRA cells, whereas granzyme B and T-bet exhibited a gradient expression. Additionally, molecules such as Runx3 and perforin were only highly expressed by a proportion of GPR56⁺ TERMA cells. Thus, not all GPR56⁺ TEMRA cells are "created equal"; instead, GPR56⁺ TEMRA cells may be further divided into phenotypically and functionally distinct subsets. Moreover, the abundance of GPR56⁺ TEMRA cells are positively correlated with the frequency of total CD4 TEMRA cells, supporting the notion that the expansion of GPR56⁺ TEMRA cells contributes to the increase in the overall magnitude of CD4 TEMRA cell responses.

The variation in CD4 TEMRA cell frequency between individuals does not appear to be a strict function of the DENV infection history, since non-DENV-infected individuals from San Diego also showed a broad range of TEMRA frequency and the two TEMRA subtypes. In some cases, TEMRA cells resembled TEM cells, while for others a clearly distinct TEMRA phenotype might be present. We speculated that this might be accounted for by other infections. We tried to address this by determining the CMV and EBV status of the analyzed donors, but this analysis was not informative as serum samples from San Diego donors were not available and nearly all Sri Lankan donors were seropositive for both viruses.

In addition to effector molecules, we found that GPR56⁺ TEMRA cells display increased expression of a set of transcription factors, including T-bet and Runx3, while downregulating the expression of ThPOK. Runx3 counteracts ThPOK and promotes the differentiation of cytotoxic CD4 T-cell, and T-bet is another key regulator that fosters the differentiation of cytotoxic CD4 T cells[14]. Additionally, we discovered that CD4 GPR56⁺ TEMRA cells upregulate the expression of the transcriptional regulator Hobit at the protein level, which has been shown to be expressed by human effector-phenotype CD8 T cells[20] as well as human cytotoxic CD4 T cells[19]. Thus, whether and how Hobit regulates the expression of cytotoxic molecules and the differentiation of GPR56⁺ TEMRA cells warrant future investigation. Moreover, Hobit directs the development of CD8 tissue-resident memory T (TRM) cells in mice[23]. Similar to their CD8 TRM counterparts, CD4 TRM cells have been shown to reside in sites of pathogen entry and are crucial for the control of various pathogens, including influenza virus and herpes simplex virus[24,25]. Since we found that GPR56⁺ TERMA cells had differential expression of molecules that mediate T-cell migration such as CCR6, it would be interesting to investigate whether CD4 TRM cells are derived from CD4 TEMRA subsets and whether Hobit regulates the establishment of CD4 TRM cells in humans.

On the basis of our data, we speculate that CD4 TEMRA cells may display a spectrum of differentiation states. At one end of the spectrum are TEMRA cells that are similar to TEM cells, as observed in group 1, and at the other end are TEMRA cells that have distinct phenotypic with enhanced expression of molecules associated with terminal differentiation and cytotoxicity such as GPR56, granzyme B, perforin, CD244, and KLRG1, as observed in group 2 donors. Thus, those GPR56⁺ TEMRA cells may provide protective immunity against viral infections by killing infected cells.

CD4 T cells with cytotoxic functions have been reported during a wide range of infections, including HIV, CMV, EBV, mouse CMV, acute lymphocytic choriomeningitis virus, influenza virus, and ectromelia virus infections[14]. DENV-specific cytotoxic CD4 T cells were initially described using T-cell clones in vitro[26] and subsequent studies on DENV-infected children further indicate that CD107a-expressing cytotoxic CD4 T cells may be associated with protection from severe dengue diseases[27]. Interestingly, previous studies from our laboratory demonstrate that CD4 TEMRA cells expand with repeated DENV infections in individuals expressing protective HLA alleles against severe dengue diseases and express enhanced levels of CD107a as well as other cytotoxic molecules such as granzyme B and perforin[9]. Here, we further characterized the CD4 TEMRA population and demonstrate that GPR56⁺ TEMRA cells, instead of GPR56⁻ TEMRA cells, had enhanced expression of cytotoxic molecules and may be correlated with HLA-associated protection against DENV. Indeed, we found that stimulated IFN-γ-producing CD4 TEMRA cells in donors with secondary DENV infections displayed a predominant GPR56⁺ phenotype, further supporting the notion that CD4 TEMRA subsets with cytotoxic functions may protect the host from severe dengue diseases.

In summary, our data demonstrate that CD4 TEMRA cells, especially GPR56⁺ TEMRA cells, are heterogeneous and display differential expression of various effector molecules and transcriptional regulators. Furthermore, CD4 GPR56⁺ TEMRA cells with cytotoxic potentials may have an important function in eliminating infected cells and may be highly relevant in vaccine-elicited protection against infectious diseases. Overall, these findings reveal immune signatures of CD4 TEMRA cells and provide new insights into the antiviral immunity against DENV and other viral pathogens.

## Methods

**Study design.** The aims of this study were to characterize the gene expression profiles, phenotypic attributes, and TCR repertoire diversity of CD4 TEMRA cells by RNA-sequencing, fluorescence-based flow cytometry, CyTOF, and TCR sequencing. All donors were screened to ensure that they had no history of anemia, HIV/HBV/HCV infections, or presence of significant systemic diseases. The Sri Lankan samples from healthy adult blood donors of both sexes and between the ages of 18 and 65 were collected anonymously by the National Blood Center, Ministry of Health, Colombo, Sri Lanka, between the years of 2010 and 2016, and processed at the Genetech Research Institute as previously described[9]. DENV-specific IgG ELISAs were performed to determine previous exposure to DENV. Flow cytometry-based neutralization assays were performed for further characterization of seropositve donors, as previously described[28]. Please note that Sri Lankan blood samples were discarded buffy coats from routine blood donations at the National Blood Center and thus exempt from human subject review as suggested by the Institutional Review Board (IRB).

Blood samples from San Diego donors were collected at La Jolla Institute for Allergy and Immunology (LJI), San Diego, California between 2010 and 2011. Prior to blood donations, all participants were screened to ensure they met inclusion criteria (18–65 males; no history of anemia, HIV/HBV/HCV infections, or presence of significant systemic diseases). Additionally, background donor information such as age and gender were collected. Peripheral blood mononuclear cells (PMBCs) isolation was performed as previously described[29]. The institutional review boards of both LJI and the Medical Faculty, University of Colombo (serving as the NIH-approved Institutional Review Board for Genetech) approved all protocols described in this study. The details of donors used in this study are listed in Supplementary Table 1.

**Serology.** DENV seropositivity was determined by anti-DENV IgG ELISA as previously described[30]. Seropositive donors that have experienced multiple infections with more than one DENV serotypes, as determined by flow cytometry-based neutralization assay[28], are referred to as secondary infections. CMV and EBV seropositivity was determined using anti-CMV IgG (GenWay Biotech, catalog# GWB-D0BE0D) and anti-EBV-VCA (Abcam, catalog# ab108730) ELISA kits, respectively, according to the manufacturer's instructions.

**Cell sorting for RNA sequencing.** PBMCs were stained with anti-human CD3, CD4, CD8, CD14, CD19, CD45RA, and CCR7 (see Supplementary Table 2 for antibody details). Subsequently, CD4 naive (CD14⁻CD19⁻CD3⁺CD8⁻CD4⁺CD45RA⁺CCR7⁺), TCM (CD14⁻CD19⁻CD3⁺CD8⁻CD4⁺CD45RA⁻CCR7⁺), TEM (CD14⁻CD19⁻CD3⁺CD8⁻CD4⁺CD45RA⁻CCR7⁻), and TEMRA (CD14⁻CD19⁻CD3⁺CD8⁻CD4⁺CD45RA⁺CCR7⁻) cells were sorted into TRIzol LS Reagent (Invitrogen) using a BD FACSAria cell sorter (BD Biosciences).

**RNA sequencing**. Total RNA was purified using a miRNeasy micro kit (Qiagen, catalog# 217084) and quantified, as previously described[31,32]. Purified total RNA (5 ng) was amplified following the Smart-seq2 protocol[33]. cDNA was purified using AMPure XP beads (1:1 ratio; Beckman Coulter). From this step, 1 ng cDNA was used to prepare a standard Nextera XT sequencing library (Nextera XT DNA library preparation kit (catalog# FC-131-1096) and index kit (set B and C, catalog# FC-131-2002 and FC-131-2003, respectively), Illumina). Samples were sequenced using a HiSeq2500 (Illumina) to obtain 50-bp single-end reads. Both whole-transcriptome amplification and sequencing library preparations were performed in a 96-well format to reduce assay-to-assay variability. Quality control steps were included to determine total RNA quality and quantity, the optimal number of PCR pre-amplification cycles, and fragment size selection (Bioanalyzer, Agilent). Samples that failed quality control were eliminated from further downstream steps. Barcoded Illumina sequencing libraries (Nextera, Illumina) were generated utilizing the automated platform (Biomek FXp). Libraries were sequenced on the HiSeq2500 Illumina platform to obtain 50-bp single-end reads (HiSeq SBS and SR Cluster v4 kits, catalog# FC-401-4002 and GD-401-4001, respectively, Illumina), generating a total of 718 million mapped reads (median of 10.1 million mapped reads per sample).

**RNA-sequencing analysis**. The single-end reads that passed Illumina filters were filtered for reads aligning to tRNA, rRNA, adapter sequences, and spike-in controls. The reads were then aligned to UCSC hg19 reference genome using TopHat (v 1.4.1)[34]. DUST scores were calculated with PRINSEQ Lite (v 0.20.3)[35] and low-complexity reads (DUST > 4) were removed from the BAM files. The alignment results were parsed via the SAMtools[36] to generate SAM files. Read counts to each genomic feature were obtained with the htseq-count program (v 0.6.0)[37] using the "union" option. After removing absent features (zero counts in all samples), the raw counts were then imported to R/Bioconductor package DESeq2[38] to identify differentially expressed genes among samples. DESeq2 normalizes counts by dividing each column of the count table (samples) by the size factor of this column. The size factor is calculated by dividing the samples by geometric means of the genes. This brings the count values to a common scale suitable for comparison. P-values for differential expression are calculated using binomial test for differences between the base means of two conditions. These P-values are then adjusted for multiple test correction using Benjamini Hochberg algorithm[39] to control the false discovery rate. We considered genes differentially expressed between two groups of samples (TEM vs. TEMRA (group 1), TEM vs. TEMRA (group 2) and TEMRA (group 1) vs. TEMRA (group 2)) when the DESeq2 analysis resulted in an adjusted P-value of <0.05 and the absolute value of log2 fold-change in gene expression was more 0.9. PCA was performed using the plotPCA method from the DESeq2 package and considering the top 1000 variable genes. Gene expression values were shown as transcripts per million (TPM) for the comparisons among different cell types. To determine possible functional relationships among the TEMRA (group 2)-specific genes the core analysis from the Integrated Pathway Analysis (IPA) software was performed (www.ingenuity.com).

**DEXSeq analysis**. Python script dexseq_prepare_annotation.py was employed to create the non-overlapping human exon reference file. The BAM files generated in the previous mapping step were used as the inputs of dexseq_count.py script to count reads on each exon. Both count matrix and exon reference file were submitted to DEXSeq package (version 1.12.2) to test the exon usage differences between two cell types[17,18]. DEXSeq fits a generalized linear model (full model) and compare it with a smaller model (reduced model), and the donor type/cell type interaction term was also introduced in both full and reduce models, so as to gauge the differences in exon usages due to cell types, independent of donors.

**Flow cytometry**. PBMCs were stained with various combinations of the antibodies listed in Supplementary Table 2. Intracellular staining for transcription factors and cytotoxic molecules were performed after fixation and permeabilization using the FlowX FoxP3 Fixation & Permeabilization Buffer Kit (R&D Systems, catalog# FC012-100). For the analysis of cytokine production, PBMCs were stimulated with DENV[40], CMV, or EBV[41,42] CD4 T-cell peptide pools (1 μg/ml for individual peptides) in the presence of brefeldin A (GolgiPlug, BD Biosciences) for 6 h and intracellular staining for IFN-γ performed after fixation and permeabilization as previously described[40]. In certain experiments, sorted GPR56− (CD14−CD19− CD3+CD8−CD4+CD45RA+CCR7−GPR56−) and GPR56+ (CD14−CD19−CD3+ CD8−CD4+CD45RA+CCR7−GPR56+) CD4 TEMRA cells were stimulated with DENV CD4 T-cell peptide pool followed by intracellular staining for IFN-γ. Samples were acquired using an LSR II flow cytometer (BD Biosciences), and the data were analyzed using FlowJo software (FlowJo, LLC).

**CyTOF**. PBMCs were stained with the viability marker Cisplatin followed by a surface antibody cocktail (Supplemental Table 3). Subsequently, cells were fixed in PBS with 2% paraformaldehyde overnight at 4 ˚C. The following day, cells were stained with an intracellular antibody cocktail (Supplementary Table 3) after fixation and permeabilization using the Foxp3 / transcription factor staining buffer set (eBioscience). Before sample acquisition, cellular DNA was labeled with iridium

intercalator (Fluidigm). Samples were then acquired suing a Helios mass cytometer (Fluidigm).

**Cell sorting for TCR sequencing**. PBMCs were stained with anti-human CD3, CD4, CD8, CD14, CD19, CD45RA, CCR7, and GPR56 (see Supplementary Table 2 for antibody details). Subsequently, cells were washed with PBS and sorted using a BD FACSAria cell sorter (BD Biosciences). The sorted subsets were defined as effector memory (CD14−CD19−CD3+CD8−CD4+CD45RA−CCR7−), central memory (CD14−CD19−CD3+CD8−CD4+CD45RA−CCR7+), GPR56− TEMRA (CD14−CD19−CD3+CD8−CD4+CD45RA+CCR7−GPR56−) and GPR56+ TEMRA (CD14−CD19−CD3+CD8−CD4+CD45RA+CCR7−GPR56+). Sorted cells were centrifuged at 3000 r.p.m. for 10 min and the pellet was frozen at −20 °C until further processing.

**TCR sequencing**. Individual T-cell clone total cDNA was obtained from $5 \times 10^4$ – $2 \times 10^6$ cells per reaction. Genomic DNA was isolated from cell samples using DNeasy blood and tissue kit (Qiagen, catalog# 69504) according to manufacturer's guidelines. Column-bound DNA was eluted in 35–100 μl, depending on cell number in the sample (35 μl for samples with < 20,000 cells, 50 μl for samples with 20,000-$1 \times 10^6$ and 100 μl for samples > $1 \times 10^6$). DNA concentrations were quantified using Nanodrop 2000 and samples were diluted for library preparation in TAE buffer (Tris-acetate-EDTA) to normalized concentrations. The sample data were generated using the immunoSEQ assay (Adaptive Biotechnologies, Seattle, WA). The somatically rearranged [species / locus] CDR3 region was amplified from genomic DNA using a two-step, amplification bias-controlled multiplex PCR approach[43,44]. Specifically, the first PCR consists of forward and reverse amplification primers specific for every V and J gene segment, and amplifies the hyper-variable complementarity-determining region 3 (CDR3) of the immune receptor locus. The second PCR adds a proprietary barcode sequence and Illumina adapter sequences. CDR3 libraries were sequenced on an Illumina instrument according to the manufacturer's instructions.

**TCR analysis**. MiXCR v1.8.3[45] was used to identify TCR chains and their abundance using quality default parameters. Here we define a clonotype as a group of T cells with the same amino acid CDR3. The number of reads determined the abundance of a clonotype (i.e.,) in a given sample. Richness of samples (i.e. # of clonotypes) and clonality[46] as a measure of dispersion of clonotypes were used to characterize TCR repertoire diversity. The normalized clonality was defined as follows:

$$\text{Normalized Shannon entropy} = -\sum_{i=1}^{n} \frac{\text{frequency}(i) \times \log_2(\text{frequency}(i))}{\log_2(n)} \quad (1)$$

$$\text{Normalized clonality} = 1 - \text{Normalized Shannon entropy} \quad (2)$$

where frequency is the function that return the frequency of the *i*th out of *n* clonotypes of the given sample. Values approaching 1 indicate an increasingly asymmetric distribution of relative abundances of a T cells toward clonotypes revealing a shift from polyclonal T-cell subset, where every clonotype has been identified by one read to a monoclonal T-cell subset, where one dominant clonotype has been identified.

**Statistical analysis**. Two-tailed Mann–Whitney or Wilcoxon tests was used to determine statistical significance between CD4 T-cell subsets using Prism software (GraphPad, La Jolla, CA).

**Data availability**. The RNA-sequencing data have been deposited in Gene Expression Omnibus (GEO) under the accession code GSE97863 and ImmPort under the study number SDY 888. The TCR sequencing data have been deposited in Sequence Read Archive (SRA) under the study number SRP119081.

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

## Acknowledgements

We thank La Jolla Institute for Allergy and Immunology Flow Cytometry Core Facility and Next-Generation Sequencing Core Facility for services, as well as the members of the Peters and Sette laboratories for their help and critical reading of this manuscript. The FACSAria II cell sorter and the CyTOF mass cytometer were acquired through the Shared Instrumentation Grant (SIG) Program S10 RR027366-01A1 and S10 OD018499-01, respectively. This work was supported by National Institutes of Health contracts and grant HHSN27220140045C and the U19 AI118626-01 to A.S.

## Author contributions

Y.T., V.S., G.P. and J.B.: Performed the experiments and analyzed the data. M.B., J.L., Z.F., J.Z.-G., J.A.G. and B.P.: Performed computational analysis. V.S.P., G.S., S.L.R. and P.V.: Prepared RNA-seq libraries and coordinated RNA-sequencing. R.N.T., A.D.D.S., S.P., G.P. and A.W.: Collected samples and provided clinical information. Y.T., M.B., D.W., A.S. and B.P.: Designed and directed the study and wrote the manuscript.

## Additional information

**Competing interests:** The authors declare no competing financial interests.

