## [Peer Review File · Nature Communications]

Reviewers' comments:

Reviewer #1 (System biology, T cell function)(Remarks to the Author):

General:

In this manuscript, Tian et al investigated the gene expression patterns of CD4+ memory T cells re-expressing CD45RA (CD4+ TEMRA cells). Before this study, it has been reported that T cells are classified into several sub-groups, depending on the expression patterns of the CD45RA and CCR7 genes. Namely, CD45RA+CCR7+, CD45RA-CCR7+ , CD45RA-CCR7- and CD45RA+CCR7-cells are classified as naïve (TN) cells, central memory (TCM), effector memory (TEM) cells and memory re-expressing cells (TEMRA) cells, respectively. However, previous studies have mostly focused on CD45RA+ cells in CD8+ T cell populations and the role of CD4+ CD45RA+ cells still remains elusive. Interestingly, it has been also shown that population of CD45RA+ cells among CD4+ T cell are varied from 0.5% to 18% between different individuals. In this paper, the authors characterized CD4 TEMRA cells in DENV or CMV infected patients. They collected CD4+ T cells from the donors, separated them into TN, TCN, TEM and TEMRA populations and compared their expression patterns. By the Principle Component Analysis (PCA), they found that expression pattern of TEMRA cells are particularly diverse between patients. Based on the PCA results, the authors further separated those cells into NT-D (near to TEM) cells and FT-D (far to TEM) cells. Particularly, they analyzed and found that FT-D cells showed high expression levels of pivotal genes of related to cytotoxicity, such as the Perforin, Granzyme B and GPR56 genes. Interestingly, GPR56+ cellular subsets showed higher clonality. Overall, the objective of the paper is timely and important and the data presented in this paper would provide insight into hitherto uncharacterized roles of CD4+ TEMRA cells. However, there are substantial remaining concerns in the current version of the manuscript as below.

Major points:

1. Biological characterizations of the TEMRA cells, particularly for the GPR56+ cells, are insufficient. The authors should demonstrate how this cell type should contribute to the overall immune response (or memory against the second infection). How those phenotypes are related to the observed clonality should be further investigated.
2. How the donors are selected? Are they matched with gender, age, previous infections other than DENV and other environmental factors? What factors determine a given donor would have an NT-D phenotype or an FT-D phenotype?
3. I wonder how general their finding on the divergence of CD4+ TEMRA cells should be for the infections of other viruses.

Minor points:

4. In Figure 7, the authors showed that IFN-gamma producing cells were GPR56+ Perforin+, and concluded that GPR56+ TEMRA may have a protective role against the infection. However, the number of IFN-gamma producing cells appeared to be very low. Indeed, the low frequency of GPR56- cells may make it difficult to clarify whether the GPR56- TEMRA should not produce IFN-gamma. It is possible that the authors simply failed to detect IFN-gamma producing GPR56- TEMRA cells. It would substantially strengthen the paper if the authors purify the GPR56+/- TEMRA cells and examine whether the GPR56+ TEMRA cells are major contributors for producing IFN-gamma in response to DENV stimulation.
5. In Figure 7, please clarify the history of the patients regarding their infections with DENV and the timing of the sample collection.
6. Figure 4: dots for the TEMRA cells seem blank.
7. How many donors were used for Figure 4A. Did all of them show the same clinical

appearance/history?

8. Sup Figs 1b and 1c: Legends for the labels are not clearly described. If they represent donors, the relation to Table 1 should be clarified.

9. Table 3: meaning of the asterisk should be described.

10. Overall, the first half of the paper should be simplified when the authors enrich the latter part to further characterize the CPG56+/- TEMRA cells.

11. Typos: l.208; "did not reached"; l. 271. "was measure"

Reviewer #2 (T cell polarization and transcription factor regulation)(Remarks to the Author):

The manuscript by Tian et al investigates a small subset of human CD4+ memory T cells that re-express CD45RA (CD4 TEMRA). Previous studies (some of them also performed by the same group of authors) indicated that these subsets are associated with protective immunity against Dengue virus infection and that these cells have a cytotoxic phenotype. In the current study, the authors performed a comprehensive gene expression analysis of peripheral human CD4 T cell subsets using RNA-seq and compared naive (TN), central memory (TCM), effector memory (TEM) and TEMRA isolated from different cohorts of donors. While there was homogeneity in the gene expression patterns in TN, TCM and TEM among various donors, there was quite some heterogeneity among the CD4 TEMRA subsets. A further analysis revealed that CD4 TEMRA cells were made of at least two subsets that can be distinguished based on the expression of CD56. The CD56+ CD4 TEMRA subset expressed genes associated with cytotoxic CD4 T cell function and a CyTOF analysis showed that CD56+ cells can be further subdivided. In addition, the CD56+ CD4 TEMRA population showed preferential clonal expansion in comparison to the CD56- CD4 TEMRA subset. While the CD56- subset was present in all donors with similar percentages, the abundance of the CD56+ population varied a lot. Finally, re-stimulation of TEMRA cells from individual associated with secondary Dengue virus infection suggested that IFN γ producing cells are predominantly present within the clonally expanded population. Taken together, based on the data the authors demonstrate heterogeneity of CD4 TEMRA cells suggesting that they can be subdivided into distinct subsets.

The authors performed a comprehensive characterization of human CD4+ TEMRA cells using RNA-seq approaches as well as CyTOF analysis. The manuscript is well written and the data are well presented.

Comments:

(1) The authors report that CD4 TEMRA cells in NT donors have a very similar gene expression profile as TEM cells (only 4 genes differentially expressed; FC >1.87). What is the interpretation/conclusion of these data? Is it possible that TEMRA NT-D cells are an intermediate population of cells that are on the way to develop from TEM into TEMRA FT-D cells? Is anything known about the function of CD4 TEMRA FT-D cells? This should be discussed.

(2) In Figure 7, the authors gate on IFN γ producing TEMRA cells. It would be informative to show the actual percentage of IFN γ expression within the TEMRA population. How representative are the data shown in Figure 7? How many different samples/donors were analyzed?

minor issue:

line 138 - the authors write that down-regulation of HNRPLL licenses the TEMRA phenotype. The

authors have not really shown that. They observed differential expression of HNRPLL between CD4 TEM and TCM compared to TEMRA cells, but not whether the down-modulation is a requirement (i.e. "a license") for TEMRA differentiation (of course differential expression of HNRPLL will lead to changes in CD45RO vs CD45RA expression).

Reviewer #3 (T cell memory and viral immunity)(Remarks to the Author):

In this manuscript, the authors present an in-depth analysis of the circulating CD4 TEMRA (terminal effector-like) subset in humans, isolated from blood. TEMRA cells comprise a phenotypically distinct (CD45RA+CCR7-) subset of T cells with effector-like properties that are more associated with the CD8+ T cell lineage where they have been extensively characterized. The frequency of CD8+TEMRA cells increases with age and is associated with persistent viruses, and CD8+TEMRA have effector-like functions and properties. CD4+ TEMRA cells, by contrast, are not well-characterized and their frequency tends to be low (<1%) among CD4+Tcells, although this can vary, with some individuals having higher CD4+TEMRA frequencies of 10% or more. The authors of this manuscript have previously reported that an increase in the CD4+TEMRA frequency was associated with repeated Dengue virus infection and moreover, an expanded frequency of CD4+TEMRA with effector properties was associated with resistance and protection to Dengue infection. In the current study, the authors show that CD4+TEMRA exhibit a transcriptional signature distinct from the corresponding TEM (Effector-memory) subset in some individuals, but in other individuals, CD4+TEMRA resemble TEM cells transcriptionally and functionally. In individuals with distinct TEMRA profiles, these cells are clonally expanded relative to TEM cells, are present within individuals with higher TEMRA frequencies, and exhibit higher cytotoxic potential as assessed by expression of perforin, CD244 and GPR56. These distinct TEMRAs are also biased to Dengue-specific populations in individuals previously infected with Dengue.

Overall, the study provides new insights into an understudied subset that will be of broad interest to those studying human immune responses, and particularly in the context of infection. The manuscript could benefit from including additional analyses of the TCR data and providing some more discussion on the bases for the differences in CD4+TEMRA frequencies as listed below.

1. The designation of the different donors as "NT-D" (near TEM donors) and "FT-D" (far-tem donors) could perhaps be reconsidered, as the implication is that the differentiation of TEMRA subset in these groups is either far from or close to TEM cells, but there is no basis to make this conclusion. I would suggest a more generic designation (group 1 or group 2) or "TEMRA-specific", or some other way of distinguishing donors with TEMRA profiles distinct from the TEM cluster. Also, there appears to be a continuum within the PCA plots, such that some donors they designated as FT-D, appear to cluster quite close to the TEM group (See supplementary figures 1 and 3.

2. The TCR data could be built upon further. What is the clonal overlap of TEMRA with TEM (or naïve or TCM) subsets? Is there more overlap between TEMRA and TEM cells from NT-D compared to FT-D donors? Do the highly expanded clones among TEMRA cells from the FT-D donors overlap with clones within the TEM subset or are they distinct? Clonal overlap can provide important information on subset relatedness and they have all of the data—it just needs to be analyzed in this context.

3. The variation in CD4+TEMRA frequency between individuals does not appear to be a function of the Dengue infection history, since non-Dengue infected individuals from San Diego also have a broad range of TEMRA frequencies and the two TEMRA subtypes. What was the CMV/EBV status of the donors they analyzed and can they provide this information in the supplementary Table 1? (If there is serum available from the donors, this serology can be readily determined). Can the authors discuss a bit more on the bases for the different TEMRA subtypes, as in some cases, since TEMRA resemble TEM cells, this subset may not really be present in those individuals, while for others, they do generate a distinct TEMRA subset.

RESPONSE TO REVIEWERS

We thank the reviewers and editors for their time and consideration. We were pleased that our report was generally well received and found the comments to be very insightful, constructive, and helpful. We have addressed every point either by performing new experiments, conducting additional analysis, and/or updating the text of the manuscript. We are grateful for the reviewers' guidance and believe that the changes have substantially improved the report. Please note that major changes to the text are shown by **yellow highlighting** in the revised report.

Reviewer #1 (System biology, T cell function)(Remarks to the Author):

General:

In this manuscript, Tian et al investigated the gene expression patterns of CD4⁺ memory T cells re-expressing CD45RA (CD4⁺ TEMRA cells). Before this study, it has been reported that T cells are classified into several sub-groups, depending on the expression patterns of the CD45RA and CCR7 genes. Namely, CD45RA⁺CCR7⁺, CD45RA⁻CCR7⁺, CD45RA⁻CCR7⁻ and CD45RA⁺CCR7⁻ cells are classified as naïve (TN) cells, central memory (TCM), effector memory (TEM) cells and memory re-expressing cells (TEMRA) cells, respectively. However, previous studies have mostly focused on CD45RA⁺ cells in CD8⁺ T cell populations and the role of CD4⁺ CD45RA⁺ cells still remains elusive. Interestingly, it has been also shown that population of CD45RA⁺ cells among CD4⁺ T cell are varied from 0.5% to 18% between different individuals. In this paper, the authors characterized CD4 TEMRA cells in DENV or CMV infected patients. They collected CD4⁺ T cells from the donors, separated them into TN, TCN, TEM and TEMRA

populations and compared their expression patterns. By the Principle Component Analysis (PCA), they found that expression pattern of TEMRA cells are particularly diverse between patients. Based on the PCA results, the authors further separated those cells into NT-D (near to TEM) cells and FT-D (far to TEM) cells. Particularly, they analyzed and found that FT-D cells showed high expression levels of pivotal genes of related to cytotoxicity, such as the Perforin, Granzyme B and GPR56 genes. Interestingly, GPR56⁺ cellular subsets showed higher clonality. Overall, the objective of the paper is timely and important and the data presented in this paper would provide insight into hitherto uncharacterized roles of CD4⁺ TEMRA cells. However, there are substantial remaining concerns in the current version of the manuscript as below.

Major points:

1. Biological characterizations of the TEMRA cells, particularly for the GPR56⁺ cells, are insufficient. The authors should demonstrate how this cell type should contribute to the overall immune response (or memory against the second infection). How those phenotypes are related to the observed clonality should be further investigated.

In response to the reviewer's comment, in order to expand the biological characterization of GPR56⁺ cells, we have now analyzed more donors and clearly show in Fig. 7 that the majority of DENV-specific TEMRA cells were GPR56⁺. Furthermore, we have analyzed other recurring viruses including CMV- and EBV-specific CD4 TEMRA cells and show in Fig. 8 that the majority of those TEMRA cells were also GPR56⁺.

To address how TEMRA cells contribute to the overall immune response and memory against infection, we have now inserted the following paragraph on page 15 of the revised manuscript:

Finally, we found that in the case of DENV-, CMV-, and EBV-specific cells, TEMRA cells constituted only 9.8%, 9.9% and 3.0% (median values) of the total response, respectively (**Supplemental Fig. 7a**). However, we observed that the TEMRA subset constituted for 34.1%, 22.6%, and 24.2% (median values) of the DENV-, CMV-, and EBV-specific cells that are associated with a cytotoxic phenotype (GPR56⁺Perforin⁺), respectively (**Supplemental Fig. 7b**), suggesting that the TEMRA subset is a relatively minor component of the response, but associated with a specific set of biological phenotypes.

Regarding the relationship between clonality and phenotypes, we have performed further analysis using the TCR sequencing data. This TCR analysis shows that GPR56⁺ TEMRA cells are likely derived from the TEM subset, not the TCM subset. These new findings on the clonal overlap between TEMRA subtypes and the other memory cell types are now presented in Supplementary Figure 6 and described as follows on pages 12-13 of the revised report:

To further assess possible relatedness of GPR56⁺ TEMRA cells and other memory cell types, we estimated the fraction of GPR56⁺ TEMRA clonotypes that were also present in either TEM or TCM cells. Supplementary Fig. 6a shows that, on average, about 40% of the GPR56⁺ TEMRA clonotypes were present in TEM cells, although these clonotypes were rare in TEM cells, and this percentage increased to ~55% if the analysis was restricted to highly represented clonotypes that had an abundance of at least 10 cells. Moreover, the highly represented clonotypes constituted about 80% of the GPR56⁺ TEMRA cells. Note that this trend was not observed for GPR56⁻ TEMRA cells (**Supplementary Fig. 6a**). On the other hand, the overlap between GPR56⁺ TEMRA clonotypes and TCM cells was considerably lower (**Supplementary Fig. 6b**). Thus, this analysis suggests that TEM cells are more likely to be a precursor of GPR56⁺ TEMRA cells than TCM cells, and GPR56⁺ TEMRA cells may result from clonal expansion of a subset of TEM cells. We cannot exclude the possibility that additional intermediate cell subsets may exist.

2. How the donors are selected? Are they matched with gender, age, previous infections other than DENV and other environmental factors? What factors determine a given donor would have an NT-D phenotype or an FT-D phenotype?

All donors were screened to ensure that they had no history of anemia, HIV/HBV/HCV infections, or presence of significant systemic diseases. The Sri Lankan samples were derived from buffy coats obtained from anonymous blood donors at the National Blood Center, Ministry of Health, Colombo, Sri Lanka, and no information is available about the donors' gender, age, or previous infections other than previous DENV

history determined by our laboratory by serological and neutralization assays. The six donors from San Diego, USA were males and between 24 and 60 years old. We have provided the gender and age information in Supplementary Table 1 and included more detailed information about the sample collection procedure in the Methods section of the revised manuscript.

Regarding the factors that determine a given donor would have an NT-D phenotype or an FT-D phenotype (please note that NT-D and FT-D have been redesignated group 1 and group 2, respectively, according to reviewers' suggestion), we have to admit that we do not know yet. Nevertheless, to assess if the infection with other viruses could give rise to an increase in GPR56⁺ TEMRA cells like we observed in DENV, we determined the phenotype of CMV- and EBV-specific CD4 TEMRA cells. We found that, like in Dengue, the majority of CMV- and EBV-specific TEMRA cells displayed a GPR56⁺ phenotype, indicating that multiple viruses can elicit an expanded cytotoxic TEMRA population. This makes it likely that the presence of an expanded GPR56⁺ TEMRA subset might be reflective of the donor's infection history. We now present these additional experiments in Figure 8 and described the results in pages 14-15 of the revised manuscript.

3. I wonder how general their finding on the divergence of CD4+ TEMRA cells should be for the infections of other viruses.

This is an excellent point. To address this question, we conducted additional experiments to analyze CMV- and EBV-specific CD4 TEMRA cells in a cohort of donors that had not been exposed to DENV. These new findings are now presented in Figure 8 and described as follows on pages 14-15 of the revised manuscript:

To investigate whether the observed phenotype of CD4 TEMRA cells specific to Dengue was also found for other viruses characterized by multiple rounds of infection, we analyzed CMV- and EBV-specific CD4 TEMRA cells in a cohort of donors that had not been exposed to DENV. CMV- and EBV-specific CD4 TEMRA cells were detected by the production of IFN- γ following stimulation with CMV and EBV peptide pools, respectively (Fig. 8a). Intriguingly, we observed that CMV-specific and EBV-specific CD4 TEMRA cells again predominantly displayed a GPR56⁺ phenotype (Fig. 8b and c), which was similar to the results of DENV-specific CD4 TEMRA cells as shown in Fig. 7. Thus, these data indicate that the cytotoxic CD4 TEMRA phenotype is observed for T cells specific for multiple viruses, in addition to DENV.

Minor points:

4. In Figure 7, the authors showed that IFN-gamma producing cells were GPR56+ Perforin+, and concluded that GPR56+ TEMRA may have a protective role against the infection. However, the number of IFN-gamma producing cells appeared to be very low. Indeed, the low frequency of GPR56- cells may make it difficult to clarify whether the GPR56- TEMRA should not produce IFN-gamma. It is possible that the authors simply failed to detect IFN-gamma producing GPR56- TEMRA cells. It would substantially strengthen the paper if the authors purify the GPR56+/-

TEMRA cells and examine whether the GPR56⁺ TEMRA cells are major contributors for producing IFN-gamma in response to DENV stimulation.

We thank the reviewer for this excellent suggestion, and have performed the experiments proposed. We sorted GPR56⁺ and GPR56⁻ CD4 TEMRA cells from secondary DENV-infected donors and measured their production of IFN- γ following stimulation with DENV CD4 T cell megapool. Consistent with the data shown in Figure 7 of the original manuscript, we observed that a higher proportion of GPR56⁺ TEMRA cells produced IFN- γ compared with their GPR56⁻ counterparts. Thus, these data support the notion that GPR56⁺ TEMRA cells were major contributors for IFN- γ production in response to DENV. These new findings are now presented in Figure 7d and described on page 14 of the revised manuscript.

5. In Figure 7, please clarify the history of the patients regarding their infections with DENV and the timing of the sample collection.

We agree that it is important to clarify this. Sri Lankan donors with secondary DENV infections were analyzed in Figure 7. The PBMC samples were derived from buffy coats obtained from anonymous blood donors at the National Blood Center, Ministry of Health, Colombo, Sri Lanka. All donors were screened to ensure that they had no history of anemia, HIV/HBV/HCV infections, or presence of significant systemic diseases and were healthy at the time of sample collection. We determined that these donors had secondary DENV infections by DENV-specific IgG ELISAs and flow cytometry-based neutralization assays. We have indicated this important information in the result, Figure 7, and the figure legend of the revised manuscript.

6. Figure 4: dots for the TEMRA cells seem blank.

To address this, we have reformatted Figure 4 to make the dots more discernible.

7. How many donors were used for Figure 4A. Did all of them show the same clinical appearance/history?

We agree that it is important to provide this information. We analyzed 5 group 1 donors and 6 group 2 donors (please note that TEMRA(NT-D) and TEMRA(FT-D) donors are now designated group 1 and group 2 donors, respectively). Furthermore, we have added two dot plots (Figure 4b) showing the percentage of GPR56⁺Perforin⁺ and GPR56⁻Perforin⁻ subsets for each donor, and these results indicate that the observed phenotypic difference between group 1 and group 2 is consistent among these donors.

8. Sup Figs 1b and 1c: Legends for the labels are not clearly described. If they represent donors, the relation to Table 1 should be clarified.

We thank the reviewer for pointing this out. We have renamed the labels in Supplementary Figure 1b and c accordingly to make it consistent with Supplementary Table 1.

9. Table 3: meaning of the asterisk should be described.

The asterisk means that an antibody was conjugated in house. To clarify this, we have add the following statement for Supplemental Table 3:

*: Antibodies were conjugated in house using the Maxpar Antibody Labeling Kit (Fluidigm) according to the manufacturer's instructions.

10. Overall, the first half of the paper should be simplified when the authors enrich the latter part to further characterize the CPG56+/- TEMRA cells.

As suggested by the reviewer, we have reduced the first half of the original manuscript by about 350 words, and have expanded the later part by adding the sections described above.

11. Typos: l.208; “did not reached”; l. 271. “was measure”

We thank the reviewer for pointing out these typos, and have modified the text accordingly.

Reviewer #2 (T cell polarization and transcription factor regulation)(Remarks to the Author):

The manuscript by Tian et al investigates a small subset of human CD4⁺ memory T cells that re-express CD45RA (CD4 TEMRA). Previous studies (some of them also performed by the same group of authors) indicated that these subsets are associated with protective immunity against Dengue virus infection and that these cells have a cytotoxic phenotype. In the current study, the authors performed a comprehensive gene expression analysis of peripheral human CD4 T cell subsets using RNA-seq and compared naive (TN), central memory (TCM), effector memory (TEM) and TEMRA isolated from different cohorts of donors. While there was homogeneity in the gene expression patterns in TN, TCM and TEM among various donors, there was quite some heterogeneity among the CD4 TEMRA subsets. A further analysis revealed that CD4 TEMRA cells were made of at least two subsets that can be distinguished based on the expression of CD56. The CD56⁺ CD4 TEMRA subset expressed genes associated with cytotoxic CD4 T cell function and a CyTOF analysis showed that CD56⁺ cells can be further subdivided. In addition, the CD56⁺ CD4 TEMRA population showed preferential clonal expansion in comparison to the CD56⁻ CD4 TEMRA subset. While the CD56⁻ subset was present in all donors with similar percentages, the abundance of the CD56⁺ population varied a lot. Finally, re-stimulation of TEMRA cells from individual associated with secondary

Dengue virus infection suggested that IFN γ producing cells are predominantly present within the clonally expanded population. Taken together, based on the data the authors demonstrate heterogeneity of CD4 TEMRA cells suggesting that they can be subdivided into distinct subsets.

The authors performed a comprehensive characterization of human CD4⁺ TEMRA cells using RNA-seq approaches as well as CyTOF analysis. The manuscript is well written and the data are well presented.

Comments:

(1) The authors report that CD4 TEMRA cells in NT donors have a very similar gene expression profile as TEM cells (only 4 genes differentially expressed; FC >1.87). What is the interpretation/conclusion of these data? Is it possible that TEMRA NT-D cells are an intermediate population of cells that are on the way to develop from TEM into TEMRA FT-D cells? Is anything known about the function of CD4 TEMRA FT-D cells? This should be discussed.

This is an excellent point. We agree with the reviewer that CD4 TEMRA cells may display a spectrum of differentiation states. At one end of the spectrum are TEMRA cells that are similar to TEM cells, as observed in NT-D (redesignated as group 1 according to reviewers' suggestion), and at the other end are TEMRA cells that have distinct phenotypic with enhanced expression of molecules associated with terminal differentiation and cytotoxicity such as GPR56, granzyme B, perforin, CD244, and KLRG1, as observed in FT-D (group 2) donors. Thus, those GPR56⁺ TEMRA cells may provide protective immunity against viral infections by killing infected cells. We have included these considerations in the Discussion section of the revised manuscript.

(2) In Figure 7, the authors gate on IFN γ producing TEMRA cells. It would be informative to show the actual percentage of IFN γ expression within the TEMRA population. How representative are the data shown in Figure 7? How many different samples/donors were analyzed?

We agree that it is important to show the frequency of IFN- γ -producing CD4 TEMRA cells as well as the consistency of the results. We analyzed a total of 10 donors, and we have generated two new panels in Figure 7, including Figure 7a and Figure 7c. In Figure 7a we show the frequencies of IFN- γ ⁺ TEMRA cells, which ranged from 0.1% to 1.9%. Figure 7c presents the composite results of Figure 7b and shows that the frequencies of total GPR56⁺, as well as GPR56⁺CD244⁺ and GPR56⁺Perforin⁺ cells were significantly higher among DENV-specific (IFN- γ ⁺) TEMRA cells compared with total TEMRA cells.

minor issue:

line 138 - the authors write that down-regulation of HNRPLL licenses the TEMRA phenotype. The authors have not really shown that. They observed differential expression of HNRPLL between CD4 TEM and TCM compared to TEMRA cells,

but not whether the down-modulation is a requirement (i.e. “a license”) for TEMRA differentiation (of course differential expression of HNRPLL will lead to changes in CD45RO vs CD45RA expression).

We agree with the reviewer that we do not have enough evidence to show that HNRPLL permits the differentiation of TEMRA cells. Therefore, we have removed the specific sentence from the revised manuscript.

Reviewer #3 (T cell memory and viral immunity)(Remarks to the Author):

In this manuscript, the authors present an in-depth analysis of the circulating CD4 TEMRA (terminal effector-like) subset in humans, isolated from blood. TEMRA cells comprise a phenotypically distinct (CD45RA+CCR7-) subset of T cells with effector-like properties that are more associated with the CD8+ T cell lineage where they have been extensively characterized. The frequency of CD8+TEMRA cells increases with age and is associated with persistent viruses, and CD8+TEMRA have effector-like functions and properties. CD4+ TEMRA cells, by contrast, are not well-characterized and their frequency tends to be low (<1%) among CD4+Tcells, although this can vary, with some individuals having higher CD4+TEMRA frequencies of 10% or more. The authors of this manuscript have previously reported that an increase in the CD4+TEMRA frequency was associated with repeated Dengue virus infection and moreover, an expanded frequency of CD4+TEMRA with effector properties was associated with resistance and protection to Dengue infection. In the current study, the authors show that CD4+TEMRA exhibit a transcriptional signature distinct from the corresponding TEM (Effector-memory) subset in some individuals, but in other individuals, CD4+TEMRA resemble TEM cells transcriptionally and functionally. In individuals with distinct TEMRA profiles, these cells are clonally expanded relative to TEM cells, are present within individuals with higher TEMRA frequencies, and exhibit higher cytotoxic potential as assessed by expression of perforin, CD244 and GPR56. These distinct TEMRAs are also biased to Dengue-specific populations in individuals previously infected with Dengue.

Overall, the study provides new insights into an understudied subset that will be of broad interest to those studying human immune responses, and particularly in the context of infection. The manuscript could benefit from including additional analyses of the TCR data and providing some more discussion on the bases for the differences in CD4+TEMRA frequencies as listed below.

1. The designation of the different donors as “NT-D” (near TEM donors) and “FT-D” (far-tem donors) could perhaps be reconsidered, as the implication is that the differentiation of TEMRA subset in these groups is either far from or close to TEM cells, but there is no basis to make this conclusion. I would suggest a more generic designation (group 1 or group 2) or “TEMRA-specific”, or some other way of distinguishing donors with TEMRA profiles distinct from the TEM cluster. Also, there appears to be a continuum within the PCA plots, such that some donors they

designated as FT-D, appear to cluster quite close to the TEM group (See supplementary figures 1 and 3).

We thank the review for this valuable suggestion. To address this, we have changed FT-D and NT-D donors to group 1 and group 2 donors throughout the manuscript.

We also agree with the reviewer that there is a continuum in the PCA plots. In order to use an unbiased criterion to group the donors according to their TEMRA phenotype, we performed hierarchical clustering, which confirms the two major groups (group 1 and group 2) identified in the PCA plots (please see Supplementary Figures 1b).

2. The TCR data could be built upon further. What is the clonal overlap of TEMRA with TEM (or naïve or TCM) subsets? Is there more overlap between TEMRA and TEM cells from NT-D compared to FT-D donors? Do the highly expanded clones among TEMRA cells from the FT-D donors overlap with clones within the TEM subset or are they distinct? Clonal overlap can provide important information on subset relatedness and they have all of the data—it just needs to be analyzed in this context.

We appreciate the reviewer for raising this excellent point. To address this question, we have performed a further analysis using the TCR sequencing data, as described in the answer to Reviewer 1 major point 1 (see above).

3. The variation in CD4+TEMRA frequency between individuals does not appear to be a function of the Dengue infection history, since non-Dengue infected individuals from San Diego also have a broad range of TEMRA frequencies and the two TEMRA subtypes. What was the CMV/EBV status of the donors they analyzed and can they provide this information in the supplementary Table 1? (If there is serum available from the donors, this serology can be readily determined). Can the authors discuss a bit more on the bases for the different TEMRA subtypes, as in some cases, since TEMRA resemble TEM cells, this subset may not really be present in those individuals, while for others, they do generate a distinct TEMRA subset.

We thank the reviewer for pointing out this important question. Regarding the question about the CMV/EBV status of the donors, due to the sample collection protocol serum samples are not available for the donors in San Diego, USA. Serum samples were available for 16 out of the 17 Sri Lankan donors analyzed in this study, and we determined as requested the CMV and EBV status of these 16 donors. 100% (16/16) and 94% (15/16) of the tested donors were CMV- and EBV-positive, respectively. Notably, three of the Sri Lankan donors were DENV-negative with two being classified into group 1 and one being classified into group 2 (Supplemental Figure 1c), and all of these three donors were CMV-positive and EBV-positive. These new results have been provided in Supplemental Table 1 as requested by the reviewer.

Given the uniformity of EBV/CMV status in the Sri Lankan cohort, it is not possible to assess the impact of this variable on TEMRA status. Instead, to assess if the infection with other viruses could give rise to an increase in GPR56⁺ TEMRA cells like

we observed in DENV, we determined the phenotype of CMV- and EBV-specific CD4 TEMRA cells. We found that, like in Dengue, the majority of CMV- and EBV-specific TEMRA cells displayed a GPR56⁺ phenotype, indicating that multiple viruses can elicit an expanded cytotoxic TEMRA population. This makes it likely that the presence of an expanded GPR56⁺ TEMRA subset might be reflective of the donor's infection history. We have added these additional experiments as Figure 8.

As requested by the reviewer, we now expanded the discussion as follows on pages 17-18 of the revised manuscript:

The variation in CD4 TEMRA cell frequency between individuals does not appear to be a strict function of the DENV infection history, since non-DENV-infected individuals from San Diego also showed a broad range of TEMRA frequency and the two TEMRA subtypes. In some cases TEMRA cells resembled TEM cells, while for others a clearly distinct TEMRA phenotype might be present. We speculated that this might be accounted for by other infections. We tried to address this by determining the CMV and EBV status of the analyzed donors, but this analysis was not informative as serum samples from San Diego donors were not available and nearly all Sri Lankan donors were seropositive for both viruses.

REVIEWERS' COMMENTS:

Reviewer #1 (Remarks to the Author):

I believe this manuscript has been very much improved by the series of extensive analyses which were conducted by the authors for the revision. Particularly, I appreciate the in-depth analysis on the clonality and heterogeneity of the newly characterized CD4+ TEMRA population. I also appreciate their careful additional analysis of the TEMRA cells in response to other viruses, CMV and EBV. I have also taken a look at the comments from the other reviewers and the responses from the authors. I found that the criticisms which are partly overlapping with my concerns in my previous review have been also properly addressed.

Reviewer #2 (Remarks to the Author):

The authors appropriately addressed my comments.

Reviewer #3 (Remarks to the Author):

The authors have addressed all of the comments and concerns with revisions and additional analyses. I would suggest that the TCR overlap graphs be moved to the main figure, rather than where they are in Supplementary figure 6, because this is non-redundant and very interesting data that should be put forth.

RESPONSE TO REVIEWERS

We thank the reviewers and editors for their time and consideration. We have used the Reviewers' and Editor's guidance in editing our manuscript to comply with the journal's format requirements and to maximize its accessibility. We appreciate the opportunity to improve our study.

Reviewer #1 (Remarks to the Author):

I believe this manuscript has been very much improved by the series of extensive analyses which were conducted by the authors for the revision. Particularly, I appreciate the in-depth analysis on the clonality and heterogeneity of the newly characterized CD4+ TEMRA population. I also appreciate their careful additional analysis of the TEMRA cells in response to other viruses, CMV and EBV. I have also taken a look at the comments from the other reviewers and the responses from the authors. I found that the criticisms which are partly overlapping with my concerns in my previous review have been also properly addressed.

We are grateful for the reviewer's guidance and agree that the changes have substantially improved the report.

Reviewer #2 (Remarks to the Author):

The authors appropriately addressed my comments.

We are grateful that the reviewer found our responses satisfactory.

Reviewer #3 (Remarks to the Author):

The authors have addressed all of the comments and concerns with revisions and additional analyses. I would suggest that the TCR overlap graphs be moved to the main figure, rather than where they are in Supplementary figure 6, because this is non-redundant and very interesting data that should be put forth.

We are delighted that the reviewer noted that we had addressed all the comments and concerns. In response to the suggestion on the TCR overlap graphs, we have moved those graphs to the main Figure 6. We thank the reviewer for this valuable comment.